

# Total ozone trends from 1979 to 2016 derived from five merged observational datasets - the emergence into ozone recovery

Mark Weber[1], Melanie Coldewey-Egbers[2], Vitali E. Fioletov[3], Stacey M. Frith[4], Jeannette D. Wild[5,6], John P. Burrows[1], Craig S. Long[5], and Diego Loyola[2]

[1]University of Bremen, Bremen, Germany
[2]German Aerospace Center (DLR), Oberpfaffenhofen, Germany
[3]Environment and Climate Change Canada, Toronto, Canada
[4]NASA Goddard Space Flight Center, Greenbelt, MD, USA
[5]NOAA/NCEP Climate Prediction Center, College Park, MD, USA
[6]INNOVIM, Greenbelt, MD, USA

*Correspondence to:* Mark Weber (weber@uni-bremen.de)

**Abstract.**

We report on updated trends using different merged datasets from satellite and groundbased observations for the time period 1979 until 2016. Trends were determined by application of a multiple linear regression (MLR) to annual mean zonal mean data. Merged datasets used are NASA MOD V8.6 and NOAA MERGE V8.6, both based upon data from the series of SBUV

and SBUV-2 satellite instruments (1978-present) and the GTO (GOME-type Total Ozone) and GSG (GOME-SCIAMACHY-GOME2) merged datasets (1995-present) that are mainly composed of satellite data from GOME, SCIAMACHY, and GOME-2A. The fifth dataset are the monthly mean zonal mean data from ground data collected at WOUDC (World Ozone and UV Data Center). The addition of four more years of data since the last WMO Ozone Assessment (2013-2016) show that for most datasets and regions the trends since the stratospheric halogens reached maximum (∼1996 globally and ∼2000 in polar

regions) are mostly not significantly different from zero. However, for some latitudes, in particular the southern hemisphere extratropics and northern hemisphere subtropics, several datasets show small positive trends of slightly below $+1\,\%/\mathrm{decade}$ that are barely statistically significant at the $2\sigma$ uncertainty level. In the tropics only two datasets show significant trends of $+0.5$ to $+0.8$ $\%/\mathrm{decade}$, while the other show near zero trends. Positive trends since 2000 are observed over Antarctic in September, but near zero trends in October as well as in March over the Arctic. Since uncertainties due to possible drifts

between the datasets as well as from the merging procedure used in the satellite datasets or due to the low sampling of ground data are not accounted for, the retrieved trends can be only considered being at the brink of becoming significant, but there are indications that we are about to emerge into the expected recovery phase. Nevertheless, the recent trends are still considerably masked by the observed large year-to-year variability in total ozone.

## 1 Introduction

The stratospheric ozone layer protects the biosphere from harmful UV radiation. One of the important measures that regulate the amount of UV radiation reaching the surface is the total column amount of ozone or in short, total ozone, which is basically





defined by the vertical integration of the ozone number density profile. As the ozone profile peaks in the lower stratosphere, total ozone is also representative of lower stratospheric ozone (from tropopause to about 27 km). The strong decline in global total ozone observed throughout the 1980s and the discovery of the Antarctic ozone hole (Chubachi, 1984; Farman et al.,

1985; Solomon et al., 1986) raised the awareness of the need to protect the ozone layer that culminated in the 1985 Vienna Convention to take actions. The main cause for the severe ozone depletion was identified as halogen containing substances also called ozone depleting substances (ODS) that are sufficiently long-lived to reach the stratosphere, releasing halogens that destroy ozone (e.g. Solomon, 1999). The Montreal Protocol and its Amendments initiated in 1986 became a binding agreement on phasing out ozone depleting substances (ODS) that ultimately initiated a decline in stratospheric halogens about ten years

later (e.g. Anderson et al., 2000; Solomon et al., 2006)

Satellite and ground-based data revealed a dramatic total ozone column decline of about $-3\,\%/$decade to $-6\,\%/$decade (dependent on latitude) throughout the 1980s until the mid-1990s that were linked to observed ODS increases (Pawson et al., 2014, and references therein). In the northern hemisphere (NH), the lowest annual mean total column ozone levels occurred in 1993, resulting from enhanced stratospheric aerosol related ozone loss after the major volcanic eruption of Mt Pinatubo in 1991

a few years before the peak in stratospheric ODS was reached (e.g. Chehade et al., 2014). In the late 1990s, annual mean total ozone increased rapidly in the NH, faster than expected from the slow decrease in ODS as a result of measures taken according to the Montreal Protocol and its Amendments. This rapid increase in the NH (Harris et al., 2008) revealed the important role of atmospheric dynamics, notably ozone transport via the Brewer-Dobson circulation that causes large variability on inter- and intra-annual time scales (e.g. Fusco and Salby, 1999; Randel et al., 2002; Dhomse et al., 2006; Harris et al., 2008; Weber et al.,

40  2011).

Apart from the inter-annual variability, total ozone levels have remained globally stable since about the year 2000. The success of the Montreal Protocol agreement is thus undisputed as the earlier decline in total ozone was successfully stopped (Pawson et al., 2014). Since ODS levels are expected to decrease slowly at about 1/3 of the absolute rate of the earlier ODS increase, it is expected that onset of ozone recovery should be evident by some increasing levels of total ozone. There are two

possible explanations as to why this has not been observed globally yet. Positive ozone trends are too small to be detected beyond the observed variability and, secondly, ODS related ozone trends are in competition with trends due to climate feedbacks. The latter means total ozone trends are not necessarily congruent to stratospheric halogens trends, e.g. have the same ratio of trends before and after the ODS peak as ODS itself. For instance, the observed increase of upper stratospheric ozone ($\sim$2 hPa) of about 2-4 % per decade since 2000 had about equal contributions from climate change and ODS changes as deduced from

chemistry-climate models (see Fig. 2-20 and related references in Pawson et al., 2014).

Regular stratospheric ozone observations started with ground-based Dobson spectrophotometers in the mid-1920s (Dobson, 1968; Staehelin et al., 1998). The number of stations with regular Dobson spectrophotometer observations strongly increased after the International Geophysical Year (IPY) 1957/1958 (Dobson, 1968). First space-based measurements of ozone occurred in 1970 with the BUV (Backscatter UV) spectrometer. Continuous measurements from space on a daily bases did not start

before the end of 1978 with the Solar Backscatter UltraViolet (SBUV) and Total Ozone Mapping Spectrometer (TOMS) instruments (McPeters et al., 2013). Starting in 1995 the SBUV-2 and TOMS observations were complemented by the European



GOME (Global Ozone Monitoring Experiment) type instruments that in addition to ozone measure other important species (NO$_2$ and OClO) relevant for stratospheric ozone chemistry (e.g. Wagner et al., 2001; Richter et al., 2005).

Global and continuous ozone observations from space now span a time period of nearly forty years. These observations now extend about 20 years beyond the global stratospheric ODS peak occurring in approximately 1996 (or 16 years after the later ODS peak in polar regions). This is near the minimum number of years of observations that has been estimated to be required to obtain statistically significant ozone trends in absence of other competing processes contributing to long-term ozone changes (Weatherhead et al., 2000).

This paper reports on updated total ozone trends by adding four more years of data (2013-2016) compared to results presented in the last WMO ozone assessment (Pawson et al., 2014). As most satellite instruments have a limited lifetime of generally less than ten years, long-term trends can be only investigated by using merged datasets. Currently there are four different satellite datasets available, two of them rely on the series of SBUV instruments covering the period since 1979 (Frith et al., 2014; Wild and Long, 2017) and two datasets that combine the European UV nadir sounders (GOME, GOME-2, OMI, SCIAMACHY) starting in 1995 (Loyola et al., 2009; Kiesewetter et al., 2010; Weber et al., 2011; Coldewey-Egbers et al., 2015). These satellite datasets are complemented by a fifth dataset that is based upon monthly mean zonal mean total ozone data derived from groundbased UV spectrometer data, mainly Dobsons and Brewers, which are collected at the WOUDC (World Ozone and UV Database Center) at Environment and Climate Change Canada (Fioletov et al., 2002). The regression analysis applied to these data is similar to what has been described in Chehade et al. (2014) and focuses on annual mean zonal mean data.

In Section 2 the five merged datasets are briefly highlighted followed by a description of the multiple linear regression (MLR) used in the trend analysis (Section 3). Section 4 shows the results of total ozone trends in rather broad zonal bands (southern and northern hemispheric extratropics and tropics) that are commonly used for ozone profile trends (Steinbrecht et al., 2017). This will allow us to look at the consistency between lower stratospheric ozone and total ozone trends. In Section 5 latitude dependent annual mean trends are presented and discussed. Results will be also shown for selected months during polar spring as recovery of Antarctic ozone levels in September have been recently reported by Solomon et al. (2016) (Section 6). A summary and final remarks are given in Section 7.

## 2  Total ozone datasets

A total of five merged and homogenized datasets are used in this study. There are two different versions of merged datasets from the series of SBUV and SBUV-2 satellite instruments (NASA SBUV MOD V8.6 and NOAA SBUV merge V8.6) being operated continuously since the late 1970s. Two merged datasets are mainly based upon the series of European satellite spectrometers GOME, SCIAMACHY, and GOME-2A which are based upon different retrieval algorithms and slightly different merging approaches (University of Bremen GSG and ESA/DLR GTO datasets). Both datasets cover the period from 1995 until today. The fifth dataset is the monthly mean zonal mean data from the network of groundbased Brewers, Dobsons, SAOZ (Système d'Analyse par Observations Zénithales), and filter instruments collected at the World Ozone and UV Data Center



(WOUDC) (Fioletov et al., 2002). The data sources are summarized in Table 1 and the various datasets briefly described in the following subsections.

## 2.1 NASA SBUV MOD V8.6

The NASA Merged Ozone Data (MOD) time series is constructed using data from the Nimbus 7 SBUV instrument and from six NOAA SBUV-2 instruments numbered 11, 14, and 16-19 (Frith et al., 2014). The instruments are of similar design, and measurements from each are processed using the same v8.6 retrieval algorithm (Bhartia et al., 2013). The Version 8.6 data contains ozone profiles in mixing ratio on pressure levels and in Dobson units on layers. The total ozone is then provided as the sum of the layer data.

To maintain consistency over the entire time series the individual instrument records are analyzed with respect to each other and absolute calibration adjustments are applied as needed based on comparison of radiance measurements during periods of instrument overlap (DeLand et al., 2012). Data from NOAA-9 SBUV-2 and data taken as the equator crossing time of the satellite approaches the terminator are of lesser quality and are excluded from the MOD composite (DeLand et al., 2012; Kramarova et al., 2013). See Frith et al. (2014) for a detailed description of the data used in MOD.

For total ozone, differences between SBUV measurements computed during the overlap periods are typically less than the differences between any given instrument and external data sources (Labow et al., 2013; McPeters et al., 2013; Frith et al., 2014). Therefore no additional adjustments to the individual instrument measurements are applied, as the adjustments are generally smaller than the inherent instrument uncertainty. Moreover there is no physical rationale to identify one instrument as better than the others, so MOD comprises all available data. During periods of overlap, data from multiple instruments are averaged.

## 2.2 NOAA SBUV Merge V8.6

The NOAA SBUV Merge V8.6 is based on the same ozone profile data retrieved with the V8.6 retrieval algorithm as described in Section 2.1. There are many methods by which the data from the various satellites can be combined. Averaging data from all available satellites in a common period as done in NASA SBUV MOD (Section 2.1) is one method to create a combined dataset. However characteristics of the measurement (e.g. time of measurement) are lost by this averaging. Another method is to identify a representative satellite for each time period as is done in the NOAA-SBUV Merge dataset. Additionally it must be determined if the data from the individual satellites can be adjusted to improve inter-satellite consistency.

Kramarova et al. (2013) shows that SBUV Version 8.6 ozone profile data from individual satellites after a meticulous cross-instrument calibration can differ by as much as 5% in various layers of the profile from data from MLS on UARS and AURA, and SAGE II due to bias differences between the instruments and potential diurnal issues above 4 hPa. Recent studies (Wild and Long, 2017) show similar differences between NOAA-18 and NOAA-19. The NOAA-SBUV dataset incorporates some corrections to individual satellite profiles. In the later period of NOAA-16 to -19 the overlaps are long, and each satellite can be compared and adjusted directly to NOAA-18 removing the small inter-satellite biases (Wild and Long, 2017).





Strong drifts in the early satellites and poor quality of NOAA-9 and NOAA-14 data can create unphysical trends when a successive head-to-tail adjustment scheme is used in the early period (Tummon et al., 2015). The current NOAA-SBUV dataset does not adjust the Nimbus-7 or NOAA-11 data, and does not include the NOAA-9 ascending node. Only the NOAA-9 descending data is adjusted to fit between the ascending and descending nodes of NOAA-11. NOAA-14 data does not appear in the final dataset, but it is used to enable a fit of NOAA-9 descending to NOAA-11 descending where no overlap exists (Wild and Long, 2017).

The total ozone product is calculated so that it remains the sum of the adjusted profile layer data. When the resulting profiles are added, many of the profile adjustments are offset. The final total ozone product is altered by less than 1 percent, and in most cases by less than $0.5\,\%$ from the original single satellite dataset.

## 2.3 GSG

The merged GOME, SCIAMACHY and GOME-2A (GSG) total ozone timeseries (Kiesewetter et al., 2010; Weber et al., 2011, 2016) consists of total ozone data that were retrieved using the University of Bremen Weighting Function DOAS (WFDOAS) algorithm (Coldewey-Egbers et al., 2005; Weber et al., 2005). The most recent modification was in the GOME-2A data record. In the WFDOAS retrieval the change in the GOME-2A instrument function with time (De Smedt et al., 2012) was accounted for by convolving ozone cross-section data with instrument function derived from daily spectral solar observations with the same instrument. Without such a correction a drift of about $+1.5\,\%/\mathrm{decade}$ becomes apparent.

The SCIAMACHY and GOME-2A observations were successively adjusted for the apparent offsets to be continuous with the original GOME data. Biases (offsets) were determined as a function of latitude in steps of 1 degree using monthly zonal means and smoothed over ten degree latitudes. Drift corrections were not applied here.

There appears a drop of the original GOME-2 data record during the 2009-2011 period relative to SCIAMACHY, which seems to be larger than the overall bias between two datasets (see Figure 1 in Weatherhead et al. (2017)). However, the very large overlap period from 2007 until 2012 between SCIAMACHY and GOME-2A was of an advantage and no further corrections beyond the latitude dependent biases were needed to adjust GOME-2A. Due to this temporary drop in the GOME-2A data, the SCIAMACHY data became the preferred choice in the merged (GSG) dataset during the overlap period (2007-2011). On the other hand, the overlap period for SCIAMACHY and GOME was very short and less than 10 months (2002-2003).

The merged GSG data is in very good agreement with WOUDC zonal mean monthly data (update from Fioletov et al., 2002, and Section 2.5) as shown in Fig. 1 of Weatherhead et al. (2017).

## 2.4 GTO

The GOME-type Total Ozone Essential Climate Variable (GTO-ECV) data record (Coldewey-Egbers et al., 2015) has been created within the framework of the European Space Agency's Climate Change Initiative (ESA-CCI) ozone project. Observations from GOME, SCIAMACHY, OMI, and GOME-2A were combined into one single homogeneous record that covers the period from July 1995 to December 2016. The total ozone columns were retrieved using the GOME-type Direct FITting





(GODFIT) version 3 algorithm (Lerot et al., 2014). In order to correct for small remaining inter-sensor biases and temporal drifts, GOME, SCIAMACHY, and GOME-2A measurements were adjusted to OMI before merging into a cohesive record. Appropriate correction factors were determined during overlap periods as a function of latitude and time. Furthermore, special emphasis was placed on the analysis of spatio-temporal sampling differences intrinsic to the satellite data and on their impact on the merged product.

Ground-based validation using Brewer, Dobson, and UV-visible instruments has shown that the GTO-ECV level-3 data record is of the same high quality as the individual level-2 data products that constitute it. Both absolute agreement and long-term stability are excellent with respect to the ground reference for almost all latitudes (Coldewey-Egbers et al., 2015; Koukouli et al., 2015) and well within the Global Climate Observing System (GCOS) target requirements (Mason and Simmons, 2011). A small number of outliers were found mostly related to sampling differences that could not be completely eradicated.

## 2.5 WOUDC data

The WOUDC ground-based zonal mean data set (Fioletov et al., 2002) was formed from ground-based measurement by Dobson, Brewer, SAOZ instruments, and filter ozonometers available from the WOUDC. Over the polar night areas, Dobson and Brewer moon measurements as well as integrated ozonesonde profiles were used. The data were screened for erroneous and unreliable measurements. The overall performance of the ground-based network was discussed by Fioletov et al. (2008).

At the next step, ground-based measurements were compared with ozone "climatology" (monthly means for each point of the globe) estimated from Nimbus-7 Total Ozone Mapping Spectrometer (N7 TOMS) satellite data for 1978–1989. Then, for each station and for each month the deviations from the climatology were calculated, and the belt's value for a particular month was estimated as a mean of these deviations. The calculations were done for 5° latitudinal belts. In order to take into account various densities of the network across regions, the deviations of the stations were first averaged over 5° by 30° cells, and then the belt mean was calculated by averaging these first set of averages over the belts. Until this point the data in the different 5-degree belts were based on different stations (i.e. were considered independent). However, the differences between nearby belts are small. Therefore, the errors of the belt's average estimations can be reduced by using some smoothing or approximation. The zonal means were then approximated by zonal spherical functions (Legendre polynomials of cosines of the latitude) to smooth out spurious variations. The merged satellite and the WOUDC data sets were compared again recently and demonstrated good agreement (Chiou et al., 2014). Estimates based on a relatively sparse ground-based measurements, particularly in the tropics and southern hemisphere, may not always reproduce monthly zonal fluctuations well. However, seasonal (and longer) averages can be estimated with a precision comparable with satellite-based data sets ($\sim 1\,\%$).

## 2.6 Data preparation

The MLR is applied to annual mean data. In this case no corrections are needed to account for auto-regression that are evident
in monthly mean timeseries (e.g. Weatherhead et al., 1998; Dhomse et al., 2006; Vyushin et al., 2007, 2010). Annual means were calculated from the monthly mean data that were all provided as zonal means in steps of 5° latitude. Annual mean data were only included for those years where at least $80\,\%$ of months in a given year were available (10 months). The SBUV





merged data have data gaps of up to three years following the Pinatubo eruption and 1-2 years following El Chichón. Broader zonal means (e.g. for 35°N-60°N) were then calculated by area weighting the 5° annual mean values contained in the bands.

At least 80% of the 5 degree zonal bands are required to make the broadband average.

All annual mean zonal mean timeseries were corrected for possible biases between them by subtracting the 1998-2008 average from each dataset and later the mean of decadal 1998-2008 averages from all datasets were added back to each dataset. That way the original values of all timeseries are nearly preserved but the bias is reduced as it is the case when using ozone anomalies.

The bias corrected GSG and GTO datasets were both extended from 1995 back to 1979 using the bias corrected NOAA data, so that MLR was always applied to the full time period starting in 1979 for all datasets. This way one ensures that all terms other than the trend terms are determined from the full time period. The NOAA data was used here as the NASA data has larger data gaps.

## 3 Multiple linear regression

In this section the MLR equation and the various explanatory variables used are briefly summarised (Section 3.1) followed by a discussion on the various choices of trend terms, e.g. independent linear trends before and after the turnaround of the stratospheric halogen (preferred choice in this study), hockey stick, or EESC (equivalent effective stratospheric chlorine) curve (Section 3.2).

### 3.1 MLR and explanatory variables

Total ozone trends are here derived from annual mean zonal mean ozone data using the MLR equation given by

$$
\begin{aligned}
y(t) = \;& a_1 \cdot X_1(t) + b_1 \cdot X_1(t)(t_0 - t) + a_2 \cdot X_2(t) + b_2 \cdot X_2(t)(t - t_0) \\
& + \alpha_{\text{sun}} \cdot S(t) + \alpha_{\text{qbo50}} \cdot Q_{50}(t) + \alpha_{\text{qbo10}} \cdot Q_{10}(t) + \alpha_{\text{ElChichón}} \cdot A_1(t) + \alpha_{\text{Pinatubo}} \cdot A_2(t) + \alpha_{\text{ENSO}} \cdot E(t) \;, \\
& + P(t)
\end{aligned}
\tag{1}
$$

where $y(t)$ is the annual mean total ozone timeseries and $t$ the year of observations. The coefficients $b_1$ and $b_2$ are the linear trends before and after the turnaround year $t_0$ when the stratospheric halogen reached its maximum abundance. In order to make both trends independent of each other (or disjoint), two y-intercepts ($a_1$ and $a_2$) are determined. The multiplication of

the independent variable $t$ with $X_i(t)$ in the first four terms of Eq. 1 describes mathematically that the first two terms only applies to the period before and the third and fourth terms to the period after the turnaround year $t_0$. $X_1(t)$ and $X_2(t)$ are given by

$$
X_1(t) = \begin{cases} 1 & \text{if } t \leq t_0 \\ 0 & \text{if } t > t_0 \end{cases}
\tag{2}
$$





and

$$X_2(t) = \begin{cases} 0 & \text{if } t \leq t_0 \\ 1 & \text{if } t > t_0 \end{cases}, \qquad (3)$$

respectively. From the calculation of the effective equivalent stratospheric chlorine (EESC) this maximum was reached at about the year $t_0 = 1996$ (Newman et al., 2007) and some years later ($t_0 \sim 2000$) in the polar regions (Newman et al., 2006, 2007).

Other main factors contributing to ozone variability and included in the MLR are the quasi-biennial oscillation (QBO), 11-year solar cycle, El Nino/Southern Oscillation (ENSO), and volcanic aerosol. The use of QBO terms (50 hPa and 10 hPa) allows a phase shift in the quasi-cyclic variation of total ozone with respect to QBO variations. The contributions from the 11-year solar cycle and QBO are in common use in total ozone MLR (e.g. Staehelin et al., 2001; Reinsel et al., 2005).

Aerosol terms related to the major volcanic eruptions like El Chichón (1982) and Mt. Pinatubo (1991) are important, in particular, to describe the deep total ozone minimum observed in the early 1990s. The volcanic aerosol effect from El Chichón eruption (1982) is independently treated in the MLR from the effect of the Mt. Pinatubo eruption (1991). The dynamical responses to the major volcanic events were quite different. While Mt Pinatubo lead to enhanced ozone depletion, the southern hemisphere (SH) extratropical total ozone rather increased as a result of a particular dynamics condition following the El Chichón event (Schnadt Poberaj et al., 2011; Aquila et al., 2013). For El Chichón the stratospheric aerosol optical depth (SAD) at 550 nm from Sato et al. (1993) is used as explanatory variable, while newer data from the WACCM model (Mills et al., 2016) is used for the period after 1990 that is dominated by the Mt. Pinatubo major volcanic eruption and also covers the series of more minor volcanic eruptions from the last decade. Though smaller, these eruptions injected sufficient amounts of aerosols into the stratosphere to affect Antarctic ozone (Solomon et al., 2016; Ivy et al., 2017). The SAD from Sato et al. (1993) is derived from satellite observation and are column amounts that extends down to about 15 km. The same data from the WACCM model represents the column amount down to the tropopause and may differ significantly from the former. The WACCM data are only available for the period after 1990 (Mills et al., 2016) and is used for the "Pinatubo" term, while for the period before 1990 the Sato et al. (1993) SAD is used in the "El Chichon" term of the MLR equation.

In the SBUV data records there are for some years not sufficient months and/or 5° latitude bands available and no annual means are calculated. If annual means of the years 1982 and 1983 are missing, the "El Chichon" term is not used in the MLR, similarly if missing all years from 1991 to 1994, the "Pinatubo" term is excluded in the MLR.

The MLR equation without the $P(t)$ term (Eq. 1) is considered the standard MLR that is commonly applied for determining trends from ozone profile data (e.g. Bourassa et al., 2014, 2017; Harris et al., 2015; Tummon et al., 2015; Sofieva et al., 2017; Steinbrecht et al., 2017). The extra term $P(t)$ in Eq. 1 accounts for other factors of dynamic variability that have been used in different combinations and definitions (e.g. accumulated, time-lagged) in the past. It includes contributions from the Arctic Oscillation (AO), and the Brewer-Dobson circulation (BDC) (e.g. Reinsel et al., 2005; Mäder et al., 2007; Chehade et al.,




2014), The BDC terms are usually described by the eddy heat flux at 100 hPa that is considered a main driver of the BDC (Fusco and Salby, 1999; Randel et al., 2002; Weber et al., 2011). The additional term $P(t)$ can be described as follows:

$$P(t) = \alpha_{\text{AO}} \cdot AO(t) + \alpha_{\text{BDCn}} \cdot BDCn(t) + \alpha_{\text{BDCs}} \cdot BDCs(t). \tag{4}$$

There are different terms for BDC in each hemisphere indicated by indices s (SH) and n (NH). The eddy heat flux is derived from daily ECMWF ERA Interim (ERA-I) reanalysis data (Dee et al., 2011). For each day the area weighted mean of the
100 hPa eddy heat flux between 45° and 75 ° latitudes is calculated and the monthly mean timeseries derived from (Weber et al., 2011). In the MLR applied to annual mean data, the winter averaged eddy heat flux is used as independent variable. The winter average, $BDCn(t)$ and $BDC_s(t)$, is derived by taking the mean from September of the previous year until April in the NH and from March until October in the SH, respectively, if not stated otherwise. For all other terms annual mean proxy timeseries are used in the MLR.

Not all terms of $P(t)$ are used in the regression since they are not entirely uncorrelated (see for instance Mäder et al., 2010; Weber et al., 2011; Chehade et al., 2014). Individual terms in Eq. 4 are only retained in the regression if the absolute value of the coefficient exceeds its $2\sigma$ uncertainty and remains robust for any combinations of terms from Eq. 4. For example the AAO term is not robust as its significance strongly depends on the BDC-S term added or not. Without the use of some additional terms contained in Eq. 4, the MLR is not able to model the large excursions in some years, e.g. 2002 in the SH or 2011 in the
NH extratropics.

The various explanatory variables and the sources of proxy time series are summarised in Table 2.

**3.2 Choice of trend terms**

In Eq. 1 the two linear trends before and after the ODS (ozone depleting substances) turnaround time $t_0$ are not continuous and are independent from each other (Pawson et al., 2014), thus we call this approach ILT (independent linear trends). All other
terms apply to the complete time period. The earliest studies of ozone recovery looked at the statistical significance of the trend after $t_0$ relative to the trend before $t_0$. The initial trend and trend change term are frequently called hockey-stick or piecewise linear trends (PLT) (Harris et al., 2008) and is mathematically equivalent to Eq. 1 without the second y-intercept or $a_2 = 0$. Several studies showed that the total ozone trend change in the extratropics is statistically significant (e.g. Reinsel et al., 2005; Harris et al., 2008; Steinbrecht et al., 2011; Mäder et al., 2010; Nair et al., 2013; Chehade et al., 2014; Zvyagintsev et al., 2015)
and this fact is considered proof that the Montreal Protocol and Amendments phasing out ODS has been working (Pawson et al., 2014).

The third possible choice is the use of the EESC curve replacing the linear regression terms (Harris et al., 2008; Mäder et al., 2010; Frossard et al., 2013; Nair et al., 2013; Chehade et al., 2014; Zvyagintsev et al., 2015). The drawback is that the long-term trend (from the fitted EESC curve) after the ODS turnaround $t_0$ is fixed relative to the trend before . The EESC
or stratospheric halogen curve indicates that the expected recovery rate in the extratropics is about one third of the absolute declining rate before $t_0$. Since the post-ODS trend is smaller, the EESC fit will be mainly determined by the fit in the declining



phase before $t_0$ and thus provides little information on trends after the ODS peak (for illustration see Fig. 1 and Kuttippurath et al. (2015)). Since the EESC as well as the linear trend terms (ILT, PLT) are the only "low" frequency terms in the MLR (while others are more or less cyclic or spiky (aerosols)), any low frequency contributions to ozone changes other than ODS

will be also fitted by these terms. In the upper stratosphere climate change and ODS roughly equally contributed to the recent ozone increase (Pawson et al., 2014), so there is no reason to assume that pre- and post-ODS trends are tied as implied by fitting the EESC.

Regardless of the use of trend terms (ILT, PLT, or EESC) the question arises when do we see the emerging of ozone recovery, i.e. ozone trends may become positive and statistically significant beyond the year-to-year variability. In this study we prefer

the use of independent linear trends (ILT) over the hockey-stick (PLT) for the following reasons. The injection point of the PLT (see Fig.1) in 1996 is quite close to the ozone minimum related to the Mt Pinatubo major volcanic eruption in 1991/92. This injection point may get lower if the aerosol effect is not properly modelled by the MLR which will likely enhance the trend after the injection point. A second important point is that the SBUV datasets have larger gaps as a result of applying a stricter filtering in the data following the major eruptions from El Chichón and Pinatubo. Enhanced aerosols interfere with the ozone

retrieval and leads to higher uncertainties (Frith et al., 2014). As a consequence the determination of the injection point of a PLT has larger uncertainties and it may affect both trends before and after $t_0$.

## 4   Trends in broad zonal bands

In Figures 2 and 3 the five bias corrected merged timeseries are shown for the extratropical $35°$-$60°$ zonal bands in the northern and southern hemisphere, respectively. In the NH the result from the MLR are only shown for the NOAA dataset and are

indicated by the orange line. In the SH the MLR results from the WOUDC data are indicated. In general the agreement between the datasets are better than with the MLR results, but also the MLR works reasonably well, explaining about $85\%$ of the variance in the timeseries. There is overall a high consistency between all datasets in the extratropics. The standard MLR plus AO and NH BDC terms were used in the NH, while in the SH only the SH BDC term was added.

Before 1997, total ozone trends in the extratropical belts between $35°$ and $60°$ in each hemisphere were about $-3\pm$

$1.5(2\sigma)\%/\mathrm{decade}$. The trends changed to about zero to $+0.5\%/\mathrm{decade}$ after the ODS peak in the extratropics. The recent trends are mostly statistically not different from a zero trend meaning total ozone levels remained stable in the extratropics over the last twenty years (1996-2016). Nevertheless, the trend change is significant and it confirms the conclusions from the last WMO ozone assessment that the ODS related decline was successfully stopped (Pawson et al., 2014).

Table 3 summarises the post-ODS peak trends for the five datasets considered here. In the NH extratropics most data show

a near zero trend. In the SH extratropics trends are positive and slightly larger than in the NH. The GSG, GTO, and WOUDC datasets indicate a positive trend of $0.7\%/\mathrm{decade}$ here barely reaching the $2\sigma$ uncertainty level. Except for the NASA dataset, all datasets show a positive trend of $+0.5\%/\mathrm{decade}$ or more in the SH.

Figures 4 and 5 (NH and SH, respectively) show how the post-ODS peak trend changed during the last decade by adding more years of observations since 2006. Up to 2010 the linear trends in the NH were at about $+1\%/\mathrm{decade}$ with an uncertainty





of just less than 2 % ($2\sigma$). With additional years after 2010 trends lowered to about $+0.5$ %/decade. The uncertainty is now reduced to slightly below 1 %. This means that a trend of 1 %/decade could be observed after 20 years of observations following the ODS peak. The below average annual mean NH total ozone in 2016 is linked to the severe Arctic ozone depletion in the same year (Manney and Lawrence, 2016) and related to the anomalous QBO induced meridional circulation changes (Tweedy et al., 2017). This resulted in a drop of the 1997-2016 NH ozone trend down to $+0.4$ %/decade (compared to $+0.6$ %/decade ending in 2015). The trend estimates are somewhat dependent on the end value in the time series. In 2010 NH extratropical ozone levels were unusually high (see Fig. 2 and Steinbrecht et al. (2011)). Despite the reasonable fitting, this high anomaly increased the trend through 2010 to $+1.8$ %/decade which was statistically significant at that time (Fig. 4).

The trend results do not vary much with additional terms used in the MLR. The standard MLR and the extended MLR (adding BDC-N and AO in the NH and BDC-S in the SH) yield about the same trend results, but the latter provides smaller uncertainties. The explained variance increases, however, significantly with the added terms ($\sim$10 % in the NH). In the SH extratropics (Fig. 5) the trends did not vary much during the last few years, but uncertainties have been reduced to slightly below $+1$ %/decade.

In the tropics both GSG and WOUDC show significant trends of $+0.8 \pm 0.4$ and $+0.5 \pm 0.5$ %/decade after 1996, respectively, while all other datasets (NASA, NOAA, GTO) show smaller and insignificant trends (Table 3 and Fig. 6). It appears that for the former datasets, in particular GSG dataset, some decadal drifts are evident. The difference between the maximum and lowest trend is less than 1%/dec. which is within the 1-3%/dec. stability requirement for long-term satellite datasets (OZONE-CCI-URD, 2016).

One should keep in mind that significance of trends in some zonal bands and for some datasets that are barely significant at $2\sigma$ can easily vanish dependent on the choice of proxies or set of fitting parameters (Chipperfield et al., 2017). Given the fact that additional uncertainties from the merging of the datasets as well as in the calculation of zonal mean data from sparse groundbased data are not accounted for here, all observed trends are likely not significant yet.

In the last ozone assessment (Pawson et al., 2014) a near global average (60°S-60°N) increase of about $+1 \pm 1.7$ % from ground and space measurements from 2000 to 2013 (corresponding roughly to a 0.8 %/decade increase) were reported. For the extended period considered here (1997-2016) the trends appear much smaller (near zero trends in the tropics and NH, except for two datasets in the tropics). Only in the SH the trends are about $0.6 \pm 0.6$ %/dec. for most datasets (see Table 3). In the extratropics trends (Figs. 4 and 5) were reduced by about half by extending the time series from 2013 to 2016, although this difference is within the trend uncertainties. It is evident from the timeseries (Figs. 2 and 3) that most of the added years since 2013 show below average ozone compared to the decade before.

The pre-ODS peak trends derived here are in good agreement with the integrated profile trends reported in Table 2-4 of Pawson et al. (2014). The trends after 1997 reported here are about half of the trends reported in the assessment, which is in line with what was discussed before. Nevertheless, within the combined uncertainties trends agree. Some of the differences may also be due to the different time periods considered (e.g. starting in 2000 versus 1997).

Our results are also largely consistent with more recent profile trend studies (Bourassa et al., 2017; Sofieva et al., 2017; Steinbrecht et al., 2017) that basically show mostly insignificant trends at lower stratosphere altitudes.



## 5 Latitude-dependent ozone trends


In Figure 7 zonal mean total ozone trends before and after the ODS peak in 1996 are shown for all five datasets as a function of latitude from 60°S to 60°N in steps of 5°. In order to better compare the results from one dataset to the others, all datasets are overplotted without their uncertainties in each panel as well. For all datasets the trends since 1996 are mostly below 1 %/decade similar to the results obtained in our previous study (Chehade et al., 2014) and what was derived from the broader

zonal bands (previous section). For some latitudes trends are barely statistically significant at $2\sigma$. Before discussing the trends in more detail, the way the MLR was applied to obtain the trends as well as some other diagnostics will be presented and discussed.

The trends were calculated using the full MLR. The regression at each latitude band was repeated by removing those terms in the extended regression (Eq. 4) for which the corresponding fit coefficient was smaller than its $2\sigma$ uncertainty. Figure 8 shows

the square correlation between the regression model and observation and $\chi$ values as a function of latitude for the NASA and NOAA regression. The square correlation varies between 0.7 and 0.9 for the full regression. The results for the NASA fit using the standard regression are also shown demonstrating that adding the BDC-S term improves the fits at SH middle latitudes and NH tropics (higher $r^2$ and lower $\chi$), while BDC-N and AO improve at NH middle latitudes. At SH low latitudes the standard model was sufficient (no additional terms needed). The importance of the BDC-S term in the NH tropics is for the first time

reported and will be discussed later.

An important question arises as to how sensitive are the trends, in particular the ones after 1996, to additional terms from Eq. 4 in the regression. As an example the trend results for the NOAA data using the standard model and the full MLR are displayed in Fig. 9. The post-ODS peak trends are nearly unchanged indicating that the recent trends are not sensitive to the additional terms used which is the case for all datasets, however, the full MLR reduces the trend uncertainty. Within the uncertainties the

pre-1996 trends are also identical in the standard and full MLR. At NH middle latitudes the addition of the BDC-NH and AO terms reduces the downward trend until 1996 by about 1 %/decade. As all proxies were not detrended, the AO and BDC-N terms also contribute to the long-term trends (thus reducing the remaining linear trends). Apart from the year-to-year variability the AO index increased throughout the 1980s along with the EESC (ODS) as shown in Fig. 1 in Weber et al. (2011) (see also Zhang et al. (2017)). The very high total ozone observed at NH middle latitudes in 2010 (Fig. 2) was linked to extreme negative

AO (Steinbrecht et al., 2011) as well as a very strong NH BDC circulation (Weber et al., 2011) during Arctic winter in the same year.

The contribution of the various factors (solar cycle, QBO, ENSO, aerosol, and so on) to ozone variability as a function of latitude is shown in Fig. 10 for two of the datasets (NASA, WOUDC). Plotted are the signed maximum responses in DU, which are the differences between the maximum and minimum value of the regression term timeseries. Negative sign means

that the ozone response is anti-correlated with the proxy change. The ozone response to the factors are in very good agreement with our previous results from Chehade et al. (2014) based upon data up to 2012. The maximum solar response of about 4-6 DU in the tropics is in agreement with the $\sim 2\%$ change from solar minimum to maximum in the lower stratosphere reported





by Soukharev and Hood (2006). Solar ozone responses are significant at all latitudes and are the result of the solar impact on atmospheric dynamics (Gray et al., 2010).

In the inner tropics the ozone response to the QBO terms changes sign poleward of 10°-15° latitudes in each hemisphere, which means positive ozone changes in the inner tropics are observed in years dominated by the QBO west phase. A new result is that the BDC-S has a significant contribution at low NH latitudes. At middle latitudes above about 40° ozone increases are associated with high absolute eddy heat fluxes (BDC proxy) as expected from the enhanced downwelling related to a stronger residual circulation. The opposite effect is seen at low latitudes (ascending branch of the BDC) with lower ozone due

to enhanced upwelling and horizontal divergence (Randel et al., 2002; Weber et al., 2011). Indeed the BDC-S ozone response has opposite signs between the low and high latitudes. The extension of the BDC-S response into NH low latitudes may be a result of the upper branch of the SH meridional circulation extending into the NH (Andrews et al., 1987) It is somewhat surprising that a similar tropical response is not evident in the NH. However, the QBO indices have a significant correlation with the BDC-N proxy ($r \sim -0.7$). The lower stratospheric QBO in the west phase (positive QBO index) allows planetary

waves to be more strongly deflected towards the equator thus reducing the perturbation of the westerly flow in the extratropical stratosphere (Baldwin et al., 2001), resulting in a weakening of the meridional winter BD circulation, lower middle latitude eddy heat flux, and reduced high latitude ozone due to reduced downwelling and higher ozone losses due to lower polar stratospheric temperatures (e.g. Weber et al., 2011).

The aerosol effect due to the Mt. Pinatubo eruption in 1991 has the largest effect on ozone at high northern latitudes with

a reduction of up to 20 DU (NASA) to 25 DU (WOUDC) in 1993. Significant ozone depletion was also observed in the NH following the El Chichón major volcanic eruption in 1982 (e.g. Hofmann and Solomon, 1989). A positive ozone response to the El Chichón is evident in the SH middle latitudes, most likely due to the specific circulation changes induced by this volcanic event (Schnadt Poberaj et al., 2011; Aquila et al., 2013). This is also believed to have caused an initial extratropical increase in SH extratropical total ozone during the first six months following the Pinatubo eruption.

Here we continue the discussion on the trend results (Fig. 7). Similar to the results from the broad zonal band trends, the latitude dependent post-ODS trends (Fig. 7) are generally smaller than the trends reported in the last ozone assessment (Pawson et al., 2014) which varied between +1 to +2 %/dec. The NH extratropical trends are below +0.5 %/dec. and statistically insignificant. In the SH trends can reach up to +0.7 %/dec. and at some latitudes barely reach the $2\sigma$ uncertainty level except for the NASA dataset.

Largest variations in trends between the datasets are seen in the tropics. Here both SBUV datasets show basically zero trends, the WOUDC and GTO negative trends in the inner tropics, and GSG positive trends that are near 10° latitudes statistically significant, reaching about +0.8 %/dec. Near the same latitudes WOUDC trends are also positive and statistically significant. One large issue is that the ground data are quite sparse in the tropics particularly at SH latitudes and generally towards the end of the data record as many stations have not submitted updates to the database yet.

An interesting result is that NH subtropical trends (20°N-30°N) peak at about +1 %/dec. and are significant with the exception of the GTO dataset which are at the lower end of the range observed. The subtropics are regions where total ozone shows quite large gradients in the transition from the tropics (lower ozone) to the extratropics (higher ozone). A shift of the





subtropical transport barrier into the tropical region could increase ozone at subtropical latitudes. Indeed a southward shift of about 5° of the tropical belt below 30 km altitude has been inferred from lower stratospheric ozone trends (Stiller et al.,

2012; Eckert et al., 2014). A recent study by Haenel et al. (2015) indicates that lower stratospheric age-of-air in the NH subtropics and extratropics has been increasing in recent years (subtropical air becoming more extratropical and reduced BDC circulation in NH), while in the SH subtropics age-of-air has variable trends in the lower stratosphere that can be negative and positive depending on altitude and is largely negative in the SH extratropics. The latter would mean that BDC circulation is getting stronger in the SH that would result in larger SH extratropical lower stratospheric ozone trends as compared to the NH.

However, the recent stratospheric ozone profile trend studies do not indicate such a hemispheric trend asymmetry in the lower stratosphere (Bourassa et al., 2017; Steinbrecht et al., 2017; Sofieva et al., 2017).

## 6   Trends in polar spring

In a recent study by Solomon et al. (2016) evidence for a significant positive trend in the SH polar region in September was reported. Other studies also indicated some early sign of ozone recovery in Antarctic spring and summer (Salby et al., 2011;

Kuttippurath and Nair, 2017). September and October are months when the ozone hole gets largest and total ozone above Antarctica reaches minimum values (see https://ozonewatch.gsfc.nasa.gov/meteorology/SH.html). A MLR has been applied to monthly mean polar total ozone for September and October in the SH as well as March in the NH. In the Arctic substantial polar ozone depletion are sporadically observed when stratospheric winter and spring are sufficiently cold (e.g. Manney et al., 2011; Manney and Lawrence, 2016). For these three months the monthly mean proxies for the respective months were used

in the MLR except for the BDC proxies which were taken as an average from March to September or October in the SH, respectively, and from September to March in the NH. We use the year 2000 as a start for the post-ODS peak trends (Newman et al., 2006). The regression results are summarised in Table 4 and MLR timeseries are shown for each of the months for one of the total ozone datasets in Fig. 11.

In SH September the post-ODS peak trends of the various datasets vary between +8 to +10 %/dec. with a $2\sigma$ uncertainty of

about 7 %/decade. In contrast the October trends are much smaller (about 3 %/dec.) and statistically insignificant. Solomon et al. (2016) and Ivy et al. (2017) showed from chemistry climate model simulation that the Calbuco volcanic event substantially contributed to the observed polar ozone loss in 2015. Even though we used the aerosol data from Mills et al. (2016), as used by Solomon et al. (2016) and Ivy et al. (2017) as input to their climate model, as a proxy in our regression, the impact of the aerosol term was found to be negligible. The Antarctic September trend is barely significant at the $2\sigma$ level. Changes in the

regression model as well a use of different proxies can easily remove the significance (de Laat et al., 2015; Chipperfield et al., 2017).

In the Arctic, March total ozone trends are quite small (below 1%/dec.) and insignificant, similar to the trends observed in NH middle latitude annual means albeit with much larger uncertainties (on the order of 4%/dec. at $2\sigma$). Also the pre-ODS peak trends in the Arctic (about −3%/dec.) are similar to the annual mean trends observed in the extratropics (30°N-60°N).

At first sight it seems surprising as in the 1990s and selected years after 2000 there was substantial polar ozone depletion. As



the polar ozone losses occur mostly in cold Arctic winters that are usually associated with years of very low BDC driving, it seems that the BDC term in the MLR accounted for the polar chemical losses. The remaining trends is in excellent agreement with the "gas phase" chemistry trends at middle latitudes (before and after the ODS peak). In the SH the polar ozone losses are much larger and the "linear" scaling of polar losses with the BDC proxy is not fully given so that the Antarctic trends are larger, or in other words the linear trends may have non-negligible contributions from polar ozone losses.

## 7  Summary and conclusions

Updated trends were derived from five different merged total ozone datasets that have been extended up to year 2016. A MLR with independent linear trends before and after the maximum stratospheric halogen content ($\sim$1996) was applied to annual means in broad zonal bands as well as narrow 5° latitude bands up to 60° latitudes. For most zonal bands and latitudes the results from the last ozone assessment (Pawson et al., 2014) and from other studies (Chehade et al., 2014; Zvyagintsev et al., 2015, and earlier studies) were confirmed that total ozone has been stable since about 1996 which is a significant change from the earlier decline observed globally outside the tropics. Globally the post-ODS peak trends vary generally between near zero trends (NH extratropics) to positive trends of $+0.7\,\%/\mathrm{dec.}$ (SH extratropics) with a statistical trend uncertainty of about $0.7\,\%/\mathrm{dec.}$ ($2\sigma$) after 20 years of observations. We may therefore conclude that we are about to emerge into the phase of ozone recovery as is predicted by ensembles of chemistry-climate models (e.g. Chipperfield et al., 2017). All post-ODS peak trends are about half of the trends reported in Pawson et al. (2014) but the changes are still within the trend uncertainties. The main reason is that in most regions total ozone in recent years showed annual means that were lower than the recent decadal mean but were well within the variability that was observed during the last twenty years.

In some regions some of the datasets show significant positive trends. In the tropical band ($< 20°$) recent trends are significant for two (GSG, WOUDC) and in the SH (35°S-60°S) for three (GSG, GTO, WOUDC) out of five datasets (Table 3). The significance of these trend estimates is close to $2\sigma$. The uncertainties reported here are purely statistical but do not account for uncertainties that may arise from the merging of the individual satellites in the merged data sets (Frith et al., 2014, 2017) as well as from sparse sampling of ground data affecting the zonal mean ground estimates. Also the significance of trends may get altered (or become insignificant) dependent on the explicit choice of regression setup (e.g. which terms to add) as well as choice of proxies for a given process.

The latitude dependent trends (Fig. 7) after 1996 are largely consistent with the results from the broader zonal bands. A striking feature is that most data sets see larger positive and statistical significant trends at subtropical latitudes between 20°N and 30°N. A southward shift of the tropical belt (e.g. Eckert et al., 2014) could be a potential explanation, however, a recent studiy shows that a markedly positive trend is not observed in most ozone profile data sets (Steinbrecht et al., 2017).

The higher trends at NH subtropics has some impact on the near global trends (60°S-60°N) derived from our MLR analyses as summarised in Table 5 and Fig. 12. Three out of the five datasets (NOAA, GSG, and WOUDC) show statistically significant trends of about $+0.6\pm0.3\,\%/\mathrm{dec.}$ on average. This trend is smaller than the trend derived from profile data for the period 2000 to 2013 ($+1.1\pm1.7\,\%/\mathrm{dec.}$) reported in Table 2-4 of Pawson et al. (2014) which was derived from the combination of ozone



profile data. Figure 12 shows the MLR results of data having the lowest (GTO) and highest post-ODS peak trends (GSG).

One should keep in mind that from MLR analyses alone we can not uniquely attribute the observed trends as they may have a significant contribution from climate change and possible feedback on atmospheric dynamics and chemistry that are difficult to disentangle without the use of chemistry-climate models.

The observed positive trends above Antarctica in September since 2000 as reported by Solomon et al. (2016) were confirmed by our MLR analysis, however, the impact from aerosols was found to be minor in contrast to the results from Solomon et al.

(2016) and Ivy et al. (2017). In October the MLR trends above Antarctica were much smaller and statistically not different from zero as were trends from the Arctic in March for all five datasets.

Adding four years of data in the various long-term total ozone data records has now further reduced the statistical uncertainties in the zonal mean trends to below $1\%/\mathrm{dec}$.. Continued ozone observations and monitoring is needed to consolidate the evidence of ozone recovery and also further improve our understanding of the complex ozone-climate feedback (in combina-

tion with climate-chemistry modelling) that will have a significant impact on future evolution of ozone (Fleming et al., 2011; Zubov et al., 2013; Pawson et al., 2014).

*Data availability.* The sources of the various datasets and proxy time series (explanatory variables) used in this study are summarised in Tables 1 and 2.

*Competing interests.* No competing interests are present.

*Acknowledgements.* M.C.E. and D.L. are grateful for the support by the ESA Climate Change Initiative project ozone_cci. M.W. and J.P.B. acknowledge the financial support of the DFG Research Unit SHARP (Stratospheric Change and its Role for Climate Prediction) and the State of Bremen.



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



**Table 1.** Start year and source of merged total ozone datasets.

| Dataset | Start year | Source |
|---|---|---|
| NASA MOD V8.6 | 1970 | http://acdb-ext.gsfc.nasa.gov/Data_services/merged/ |
| NOAA SBUV merge V8.6 | 1978 | ftp://ftp.cpc.ncep.noaa.gov/SBUV_CDR/ |
| GSG | 1995 | http://www.iup.uni-bremen.de/gome/wfdoas |
| GTO | 1995 | http://atmos.eoc.dlr.de/gome/gto-ecv.html |
| WOUDC | 1964 | http://woudc.org/archive/Projects-Campaigns/ZonalMeans/ |

**Table 2.** Sources of explanatory variables / proxy timeseries used in the MLR.

| Variable | Proxy | Source |
|---|---|---|
| $S(t)$ | Bremen composite Mg II index (Snow et al., 2014) | http://www.iup.uni-bremen.de/UVSAT/Datasets/mgii |
| $QBO_{50}(t)$, $QBO_{10}(t)$ | Singapore wind speed at 50 and 10 hPa (update from Naujokat, 1986) | http://www.geo.fu-berlin.de/met/ag/strat/produkte/qbo/qbo.dat |
| $E(t)$ | MEI (ENSO) Index (Wolter and Timlin, 2011) | https://www.esrl.noaa.gov/psd/enso/mei/ |
| $AO(t)$, $AAO(t)$ | Antarctic Oscillation (AAO), Arctic Oscillation (AO) | http://www.cpc.ncep.noaa.gov/products/precip/CWlink/daily_ao_index/teleconnections.shtml |
| $A_1(t)$ | stratospheric aerosol depth at 550nm ($t < 1990$) (update from Sato et al., 1993) | https://data.giss.nasa.gov/modelforce/strataer/tau.line_2012.12.txt |
| $A_2(t)$ | stratospheric aerosol depth at 550nm from WACCM model ($t \geq 1990$) (Mills et al., 2016) | http://dx.doi.org/10.5065/D6S180JM |





**Table 3.** 1979-1996 and 1997-2016 annual mean total ozone trends in broad zonal bands. Uncertainties are provided for $2\sigma$ and trends in bold indicate statistical significance. $r^2$ is the square Pearson correlation and $\chi$ the residual defined as $\chi^2 = \sum_i (\text{obs}_i - \text{mod}_i)^2/(n-m)$, where $\text{obs}_i$ are the observations and $\text{mod}_i$ the MLR model, $n$, the number of data (years) in the timeseries, and $m$, the number of parameters fitted. In the NH standard MLR plus AO and BDC-N terms were used; in the SH and tropics standard MLR plus SH BDC term were used.

| zonal bands | MLR | | NASA | NOAA | GSG | GTO | WOUDC |
|---|---|---|---|---|---|---|---|
| 35°N-60°N | standard + AO + BDC-N | trend >1996 [%/dec.] | +0.2(8) | +0.4(8) | +0.2(8) | −0.1(8) | +0.2(8) |
| annual | | trend ≤1996 [%/dec.] | **−2.8(15)** | **−3.1(14)** | — | — | **−2.8(15)** |
| | | $r^2$ | 0.83 | 0.85 | 0.84 | 0.85 | 0.83 |
| | | $\chi$ [DU] | 3.5 | 3.3 | 3.3 | 3.2 | 3.6 |
| 20°S-20°N | standard + BDC-S | trend >1996 [%/dec.] | +0.1(3) | +0.2(3) | **+0.8(4)** | 0.0(4) | **+0.5(5)** |
| annual | | trend ≤1996 [%/dec.] | −0.3(6) | −0.5(6) | — | — | +0.2(8) |
| | | $r^2$ | 0.87 | 0.87 | 0.85 | 0.83 | 0.77 |
| | | $\chi$ [DU] | 1.1 | 1.2 | 1.3 | 1.3 | 1.7 |
| 35°S-60°S | standard + BDC-S | trend >1996 [%/dec.] | +0.3(7) | +0.6(8) | **+0.7(7)** | **+0.6(6)** | **+0.7(7)** |
| annual | | trend ≤1996 [%/dec.] | **−3.6(14)** | **−3.4(14)** | — | — | **−3.4(13)** |
| | | $r^2$ | 0.89 | 0.89 | 0.90 | 0.91 | 0.87 |
| | | $\chi$ [DU] | 3.0 | 3.1 | 2.7 | 2.6 | 3.0 |

bold numbers: statistical significance at $2\sigma$                                    .





**Table 4.** 2000-2016 polar total ozone trends in March (NH), September (SH), and October (SH). Uncertainties are provided for $2\sigma$ and trends in bold indicate statistical significance. $r^2$ is the square Pearson correlation and $\chi$ the residual (see caption of Table 3). The results were obtained from the standard MLR with the respective hemispheric BDC term added.

| zonal bands | MLR | | NASA | NOAA | GSG | GTO | WOUDC |
|---|---|---|---|---|---|---|---|
| 60°N-90°N | standard + BDC-N | trend ≥2000 [%/dec.] | +0.4(37) | +1.2(37) | +0.9(39) | +0.5(37) | +0.4(45) |
| March | | trend <2000 [%/dec.] | −2.0(63) | −3.4(64) | — | — | −2.8(75) |
| | | $r^2$ | 0.80 | 0.81 | 0.80 | 0.80 | 0.70 |
| | | $\chi$ [DU] | 14.2 | 14.5 | 15.2 | 14.2 | 17.7 |
| 60S°S-90°S | standard + BDC-S | trend ≥2000 [%/dec.] | **+10.1(69)** | **+8.1(73)** | **+8.2(62)** | **+9.1(56)** | **+8.6(68)** |
| September | | trend <2000 [%/dec.] | **−12.2(107)** | **−13.9(114)** | — | — | **−19.3(106)** |
| | | $r^2$ | 0.82 | 0.85 | 0.90 | 0.90 | 0.88 |
| | | $\chi$ [DU] | 14.1 | 15.0 | 12.8 | 12.0 | 14.0 |
| 60°S-90°S | standard + BDC-S | trend ≥2000 [%/dec.] | +0.9(77) | +2.1(71) | 2.7(76) | +2.7(79) | +5.7(102) |
| October | | trend <2000 [%/dec.] | **−18.0(122)** | **−18.1(112)** | — | — | −12.7(161) |
| | | $r^2$ | 0.82 | 0.84 | 0.81 | 0.81 | 0.75 |
| | | $\chi$ [DU] | 16.8 | 15.5 | 16.6 | 17.2 | 22.3 |

bold numbers: statistical significance at $2\sigma$

.

**Table 5.** 1979-1996 and 1997-2016 annual and near global mean total ozone trends. For further information on variables see Table 3. Results are from the standard MLR and the full MLR including BDC terms from both hemispheres and AO term.

| zonal bands | MLR | | NASA | NOAA | GSG | GTO | WOUDC |
|---|---|---|---|---|---|---|---|
| -60°S-60°N | full | trend >1996 [%/dec.] | +0.2(3) | **+0.5(4)** | **+0.7(3)** | +0.2(3) | **+0.6(3)** |
| annual | | trend ≤1996 [%/dec.] | **−1.8(7)** | **−2.0(7)** | — | — | **−1.2(6)** |
| | | $r^2$ | 0.92 | 0.92 | 0.94 | 0.94 | 0.92 |
| | | $\chi$ [DU] | 1.3 | 1.3 | 1.2 | 1.2 | 1.2 |
| -60°S-60°N | standard | trend >1996 [%/dec.] | +0.2(3) | **+0.5(3)** | **+0.7(3)** | +0.2(3) | **+0.6(4)** |
| annual | | trend ≤1996 [%/dec.] | **−2.1(7)** | **−2.3(7)** | — | — | **−1.7(6)** |
| | | $r^2$ | 0.90 | 0.91 | 0.91 | 0.93 | 0.86 |
| | | $\chi$ [DU] | 1.3 | 1.4 | 1.3 | 1.2 | 1.4 |

bold numbers: statistical significance at $2\sigma$                                                                    .





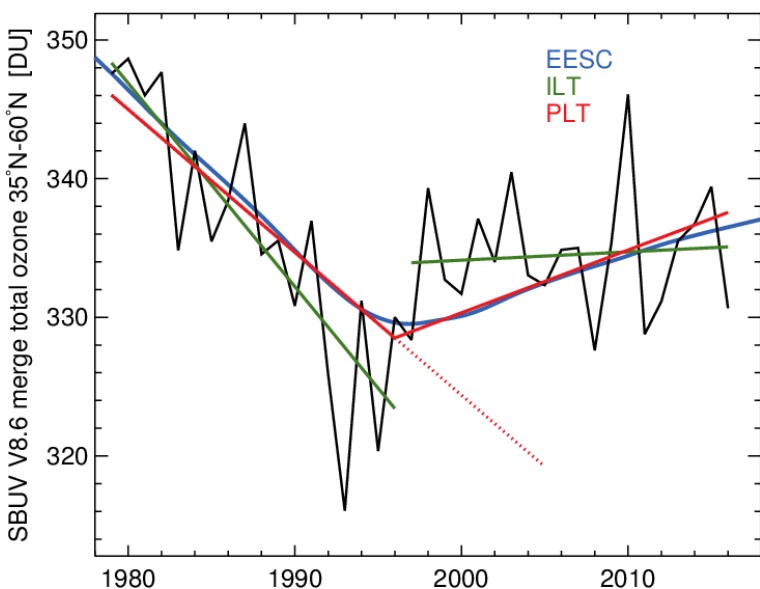

**Figure 1.** Illustration of different choices of trend terms commonly used in MLR applied to total ozone. Blue: EESC (effective equivalent stratospheric chlorine); Red: piecewise linear trends before and after $t_0 = 1996$ (PLT) also called hockey-stick; Green: independent linear trends (ILT). Black curve shows the NH total ozone timeseries from NOAA SBUV V8.6. The red dotted line indicates that the PLT is mathematically equivalent to using a trend change term in the MLR. The injection point is the point where the trend change terms starts, here in year 1996. All fits were done using only the linear regression terms in Eq.1 or, alternatively, the EESC curve replacing linear regression terms. See discussion in main text.





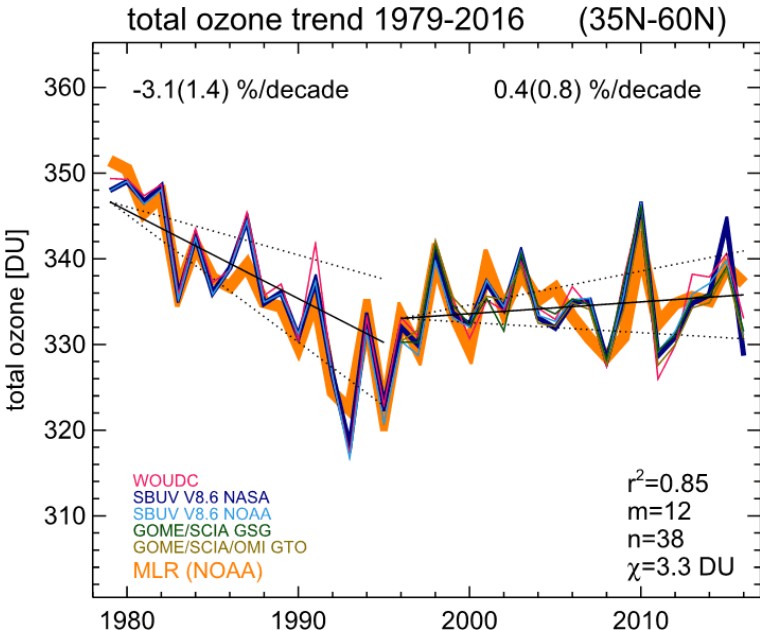

**Figure 2.** NH annual mean total ozone timeseries of five bias corrected merged datasets in the 35°N-60°N latitude band (NH extratropics). The thick orange line is the result from applying MLR Eq. 1 to the NOAA timeseries. In addition to the standard MLR, AO and BDC-N terms are included (see Eq. 4). $n$ is the number of data (years) used in the MLR and $m$ the number of parameters fitted. The square of the correlation between observations and MLR is given by $r^2$. $\chi^2$ is the sum square of the timeseries minus MLR divided by the degrees of freedom ($n-m$). The solid lines indicate the linear trends before and after the ODS peak, respectively. The dotted lines show the $2\sigma$ uncertainty of the MLR trend estimates. Trend numbers are indicated for the pre- and post-ODS peak period in the top part of the plot. Numbers in parentheses are the $2\sigma$ trend uncertainty.





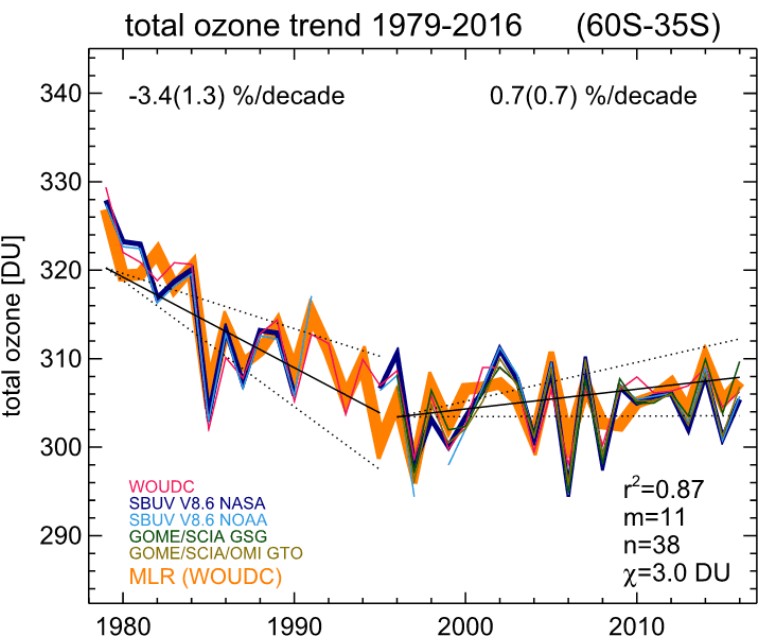

**Figure 3.** Same as Fig. 2, but for 35°S-60°S zonal band (SH extratropics) and MLR applied to WOUDC ground data. Standard MLR plus BDC-S term was applied to the WOUDC data.




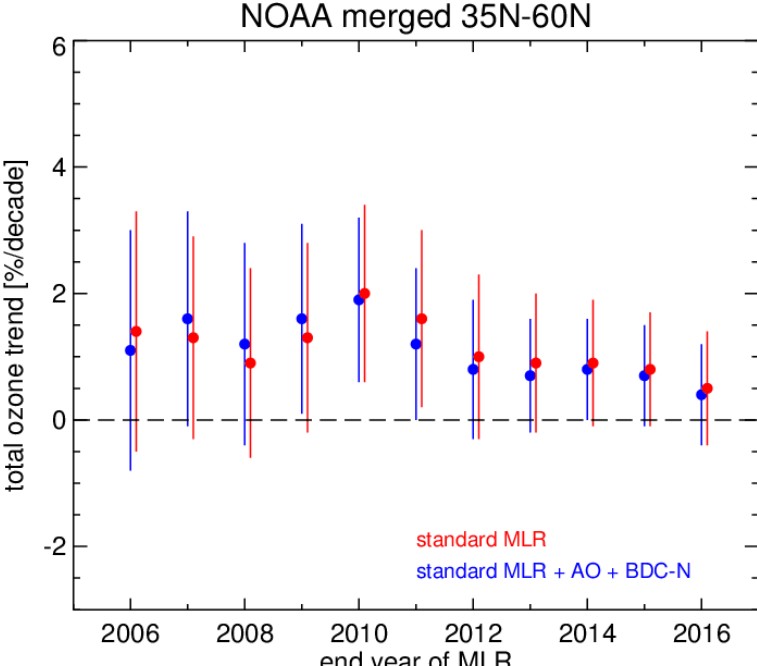

**Figure 4.** The dependence of the post-ODS peak trends in the NH extratropics from the end year in the MLR. The vertical bars show the $2\sigma$ uncertainties of the trends. Red symbols are the results from the standard MLR fit (Eq. 1 with $P(t) = 0$) and blue from the extended MLR that includes the AO and NH BDC terms (see Eq. 4).



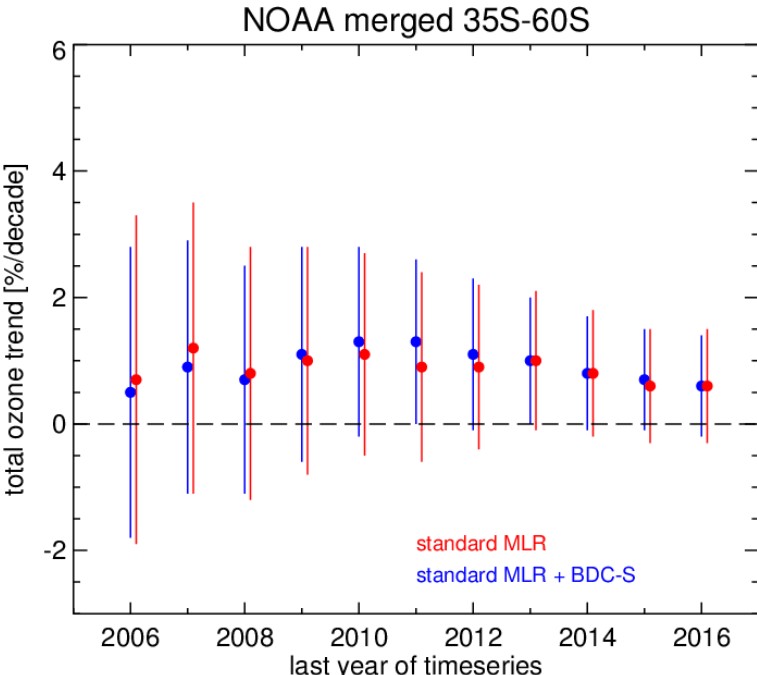

**Figure 5.** Same as Fig. 4.The vertical bars show the $2\sigma$ uncertainties of the trends.

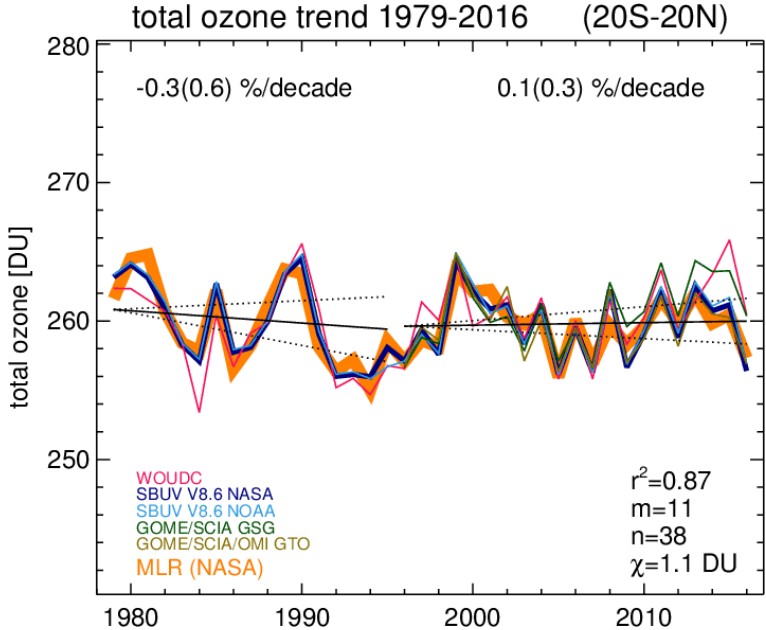

**Figure 6.** Same as Fig. 2, but for 20°S-20°N zonal band (tropics) and MLR applied to NASA SBUV Mod V8.6. In the tropics the standard MLR plus BDC-S term was used.



**Figure 7.** Linear trends and in %/dec. and $2\sigma$ uncertainty bars before (red) and after (blue) year 1996. a) NASA SBUV, b) NOAA SBUV, c) WOUDC, d) GSG, and e) GTO. Trends were calculated in 5° zonal bins from 60°S to 60°N using the full regression model. In panel d) and e) the trends before the pre-ODS peak are not shown as the GSG and GTO are mainly available after 1995. In light colors (red and blue) trends from all datasets are overlaid in each panel to facilitate comparison.





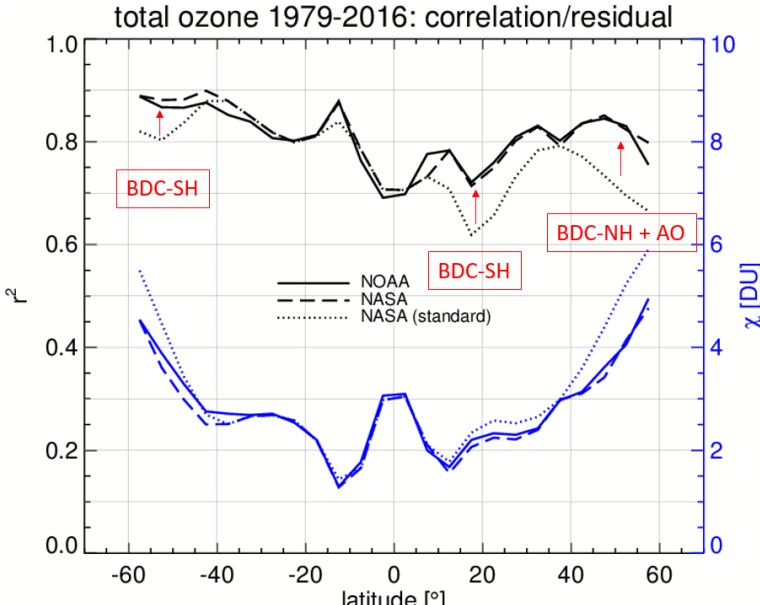

**Figure 8.** Correlation ($r^2$) between observed time series and regression (black) and MLR residual ($\chi$, blue) as a function of latitude. Results are shown for NASA and NOAA data using the full regression as well as results from standard MLR (NASA only). See caption for Fig. 2 for the definition of $\chi$. Improvement in the regression is evident from adding BDC-S at SH middle latitudes and NH subtropics and by adding BDC-N and AO terms (NH middle latitudes) to the standard regression as indicated by the red arrows.



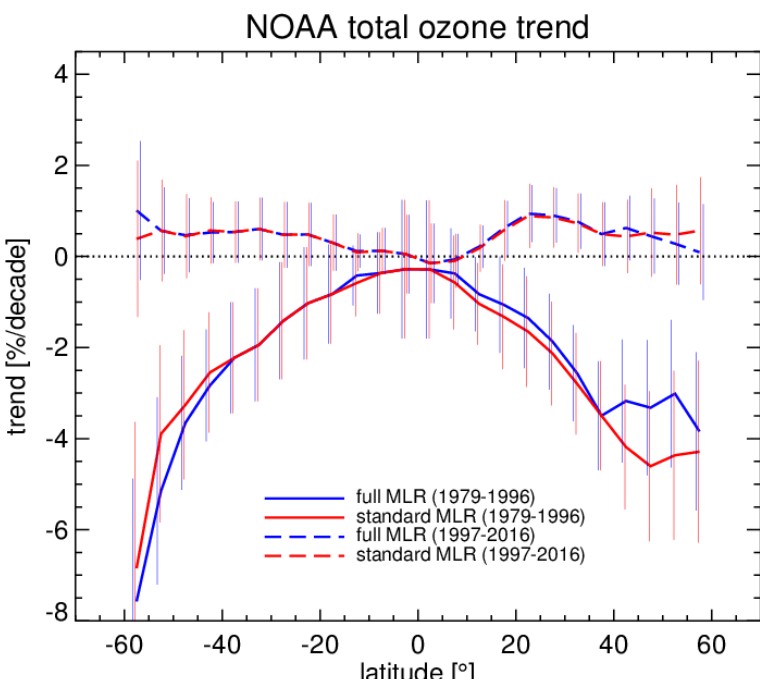

**Figure 9.** Linear trends in %/decade before and after 1996 by applying the standard (red) and extended MLR (blue) to NOAA data. Uncertainties are given as $2\sigma$. Dash lines are the trends after 1996 and solid lines before 1996.





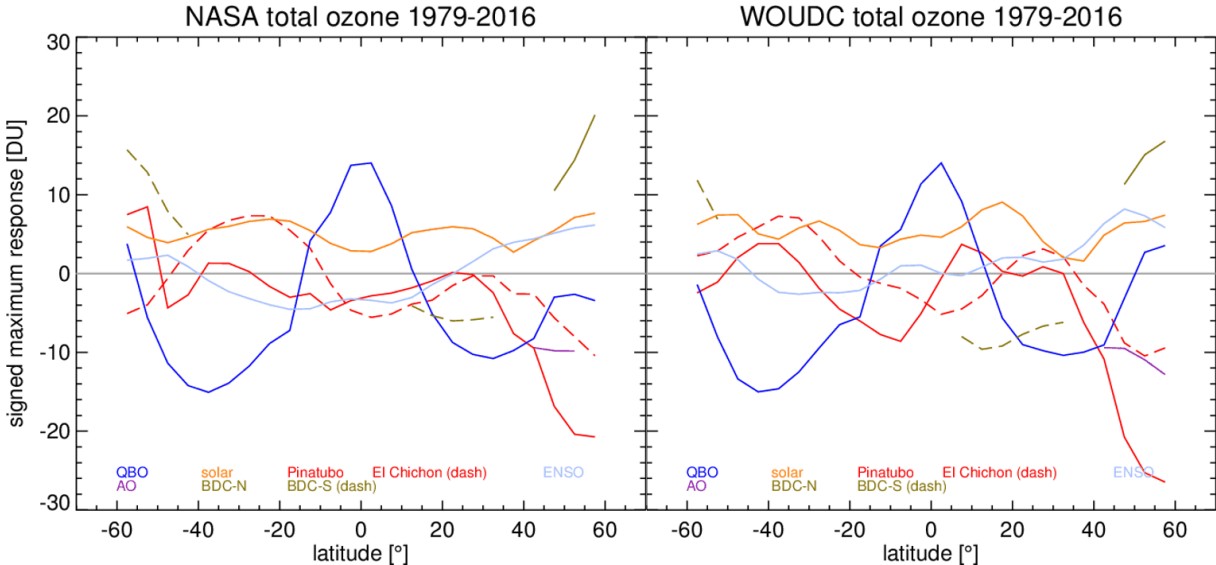

**Figure 10.** Signed maximum response during the period 1979-2016 from various factors (terms) in the MLR. Left: NASA data; right: WOUDC data. Negative values mean that total ozone is anti-correlated with the corresponding proxy (factor). Maximum response is the difference between the maximum and minimum value of the regression term in the MLR timeseries. Note: in the MLR regression negative values of the BDC-S proxy are used, meaning that positive values corresponds to enhanced BDC driving in both hemispheres.





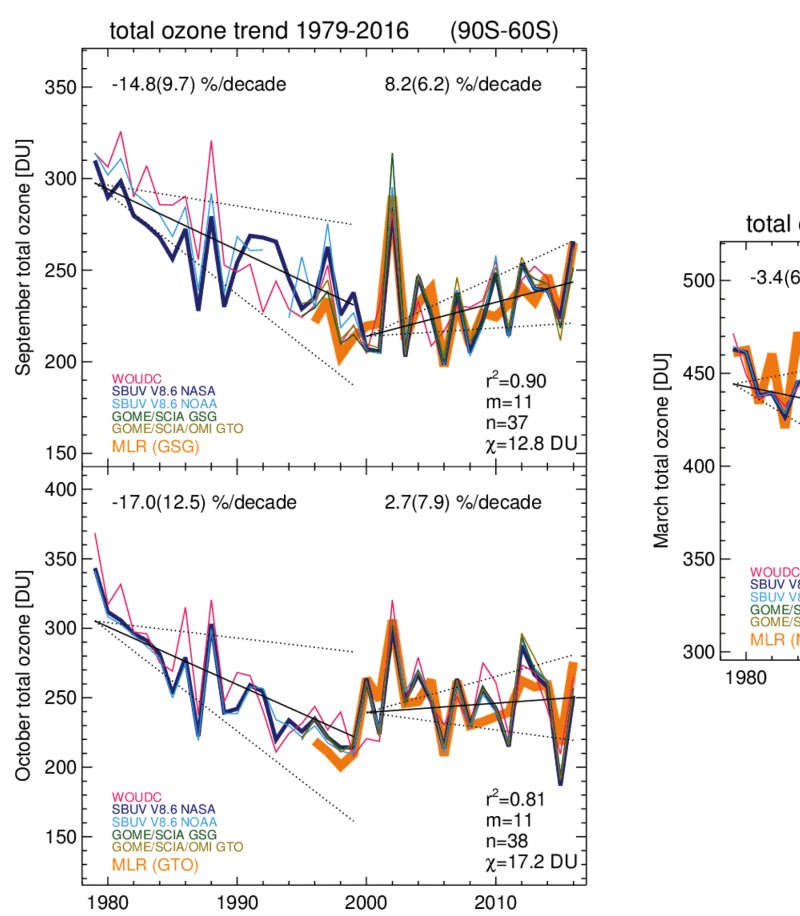

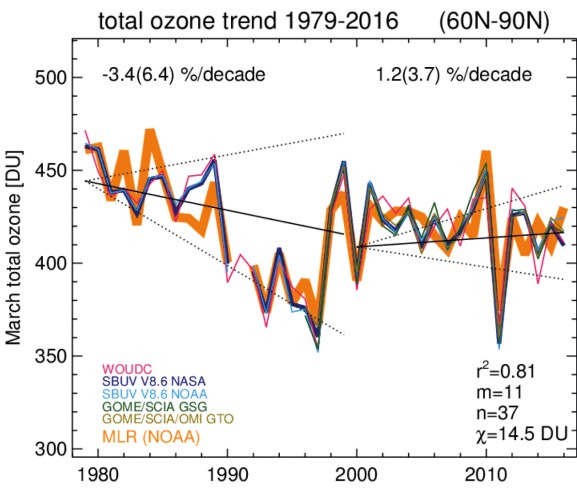

**Figure 11.** Total ozone timeseries for the SH and NH polar cap (60°-90°) and MLR timeseries (orange line) applied to one of the datasets. Left top: SH September and MLR applied to GSG; left bottom: October and MLR applied to GTO. Right: NH March and MLR applied to NOAA. MLR results are shown for the standard regression plus respective hemispheric BDC term.





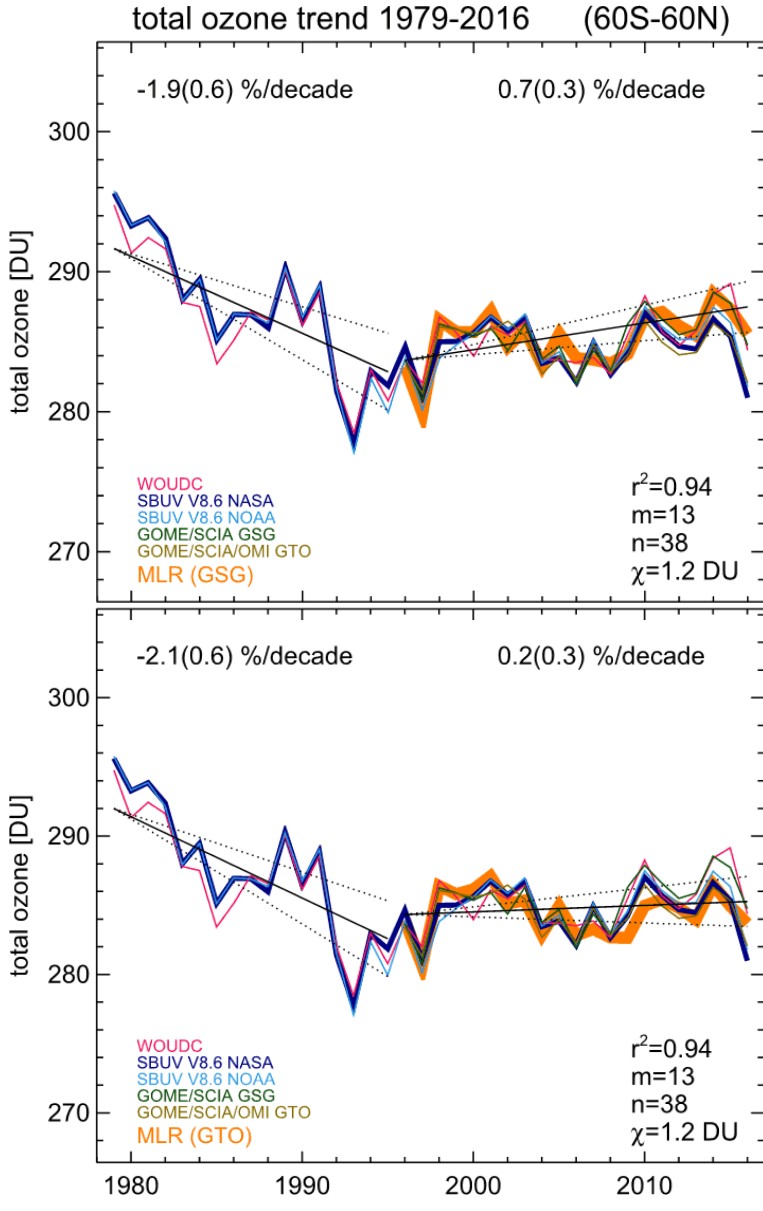

**Figure 12.** Near global total ozone timeseries (60°S-60°N) and MLR timeseries (orange line) applied to GSG (top) and GTO (bottom). Full MLR was applied including both BDC terms and AO.