# Peer review of "Total ozone trends from 1979 to 2016 derived from five merged observational datasets - the emergence into ozone recovery"

_Atmospheric Chemistry and Physics, 2017_

## Referee Comment (RC1) · Anonymous Referee #1 · 21 Oct 2017

Overall comment: This is an important paper that should be published. It's important that the excellent prior analysis of ozone trends by this group in Chehade et al. be extended with additional years and documented. I have a number of major comments and questions that I believe should be addressed, and a few minor ones.

Major comments

1) Please clarify the extent to which there are trends in any of the terms used in the MLR, whether there are uncertainties in those, and whether these in turn can influence the calculated ozone trends and their uncertainties. For example, I would argue that we

do not know the trends in eddy heat fluxes in the stratosphere very well (although we may know the year to year variability, we do not know the longer term trends on decadal or multi-decadal time scales). So one question is: are there trends in the eddy het flux terms that characterize the BDC components in the MLR? How uncertain are those trends? Could they (or do they) then influence the ozone trends that are the primary subjects of interest here? Papers on the uncertainties and differences between ERA and MERRA might be a useful point of reference here, but only a starting point; I think quantitative analysis is needed. The same could be said for trends in the solar term, for example. I am concerned that these could considerably influence the conclusions drawn, and should be discussed and documented.

2) Please clarify which terms in the MLR regression could conceivably involve feedbacks. For example, it is possible that changes in ozone play a role in the strength of the BDC (and this could happen not only on longer time scales, but also interannually). Has this been considered? Could it be important? If, for example, part of the BDC trend term is caused by ozone changes, then is your calculation of the 'ozone trend' potentially in error? By how much?

3) There are many studies providing evidence that the Antarctic ozone hole has influenced the strength of the southern annular mode (AO) in some seasons. Here is another potential feedback. The same questions apply as in item 2) above.

4) The fit to the interannual ups and downs in the ozone time series is pretty good, implying that on an interannual basis the terms involving dynamical variability are fairly well captured. Thus, the ups and downs are certainly not random noise – they are due to known and characterizable phenomena. The paper ought to discuss this, and make reference to the work of Shepherd et al., NatGeo, 2014, who combined a model with data to improve on the analysis. Based on the Shepherd paper, it does not seem reasonable to allow these variations to inflate the uncertainties on the ozone trend terms. Instead of doing the approach of combined MLR, would it not be more consistent to remove them first, and then examine the trends in the remainder. Terms involving interannual dynamical variations could conceivably be removed by detrending your index, and then regressing the detrended series to the ozone time series, and then doing MLR with the remaining terms you have. How would this or a similar approach affect the ozone trends, and in particular, their uncertainties? If this could significantly reduce the uncertainties in ozone trends (as I suspect, and as I think Shepherd et al. support), then that should at a minimum be stated since uncertainties are a key emphasis of the paper, and we need to know how robust the ozone trend uncertainties really are.

5) While the ILT and PLT trend approaches have certain advantages, they are allowed to float independent of each other. So what they lack is a grounding in the physics and chemistry that must link the processes that deplete ozone to those that make it recover. The advantage of EESC is that it has that grounding. The ILT 'advantage' over the PLT as suggested by your figure has a lot to do with the choice of year for the separation, which is a little arbitrary since there are uncertainties in EESC, and it does vary with height as well as latitude. I think the advantages and disadvantages of each approach should be discussed more clearly.

6) I'm concerned about some of the statements regarding the aerosol fits, for several reasons. a) First, it's not obvious that the total SAD is the best predictor, because the distribution of aerosol and the distribution of ozone losses need not coincide. At a minimum, the paper should test what happens if the SAD amount at 70 mb is used instead. b) Second, the statement that you used the Mills et al. aerosols but don't get significant correlation with polar ozone (suggesting a conflict with Solomon et al., 2016) misses some key points. As indicated in Solomon et al., 2016 and discussed in greater detail in Ivy et al. (GRL, 2016), the dependence of aerosols on ozone loss will depend upon temperature; in a warm year even a big volcano doesn't have a big effect so the dynamics is important in setting the stage. To capture this, you might need some kind of mixed predictor including aerosols and temperature, but that's not what you used so it's misleading to say that you don't get a high correlation. c) Third, the key parameter emphasized in Solomon et al., 2016 was the area of the ozone hole, which is not the

variable you have evaluated here.

Some minor comments

7) Page 2, line 44 what is the reference for this number for changes in ODS levels?

8) Page 3, line 73. Similar but not the same? What is different from Chehade? Please summarize what you changed, and why.

---

## Referee Comment (RC2) · Anonymous Referee #2 · 30 Oct 2017

Review for "Total ozone trends from 1979 to 2016 derived from five merged observational datasets - the emergence into ozone recovery"

General comments

Here authors use five merged data to determine ozone trends before and after ODS peak in the stratosphere. Since last few years, there are various studies claiming to have detected signs of ozone recovery. So this is a timely study showing that although stratospheric ozone is no longer decreasing, positive ozone trends in the observational data are still statistically insignificant. This is not unexpected due to large dynamical variability but ozone observations are necessary to monitor the evolution of

stratospheric ozone as well as improve our understanding of complex ozone-climate feedbacks.

Overall this is well-written manuscript and I strongly recommend it for the publication.

Minor comments:

Line 13: other–: others

line 18:Ăă Do you mean "dynamical variability"

line 22: "outside the tropics ozone profile peaks

line 164: please mention the number of outliers in percentage

Line 191-195- please mention the data sets with largest biases for few latitude bands along with absolute biases In DU.

Line 224: "deep total ozone minimum–> large ozone decrease

Line 228: confusing.. those two references are about Pinatubo eruption. Also, it is not only dynamics but vertical distribution of SAD caused increase in SH mid- stratospheric ozone (e.g. Dhomse et al., 2015)

Line 228: SAD= area density but SAOD= stratospheric aerosol optical depth

Line 229: Can't understand why you are using WACCM simulated SAD, which is still data. I think CMIP6 recommended SAD data Ăă(or AOD@550nm) until 2014 would have been much more realistic ftp://iacftp.ethz.ch/pub_read/ For remaining two years, WACCM-based or any other data would be OK.

Line 250: what about tropics? Did you try to use mean heat flux from both the hemispheres (BDCn +BDCs)/2.?

Line 280: reword "roughly equally contributed"

line 283: "emerging signs of ozone recovery"

line 289: Volcanically enhanced stratospheric aerosol

line 310: with an uncertainty just less than (delete of)

line 314: I think "Osprey et al, 2016" would be a better reference.

line 382: Again. What happens if you combine EP fluxes from both the hemispheres?

Line 397: Do you Mt. Pinatubo eruption? Or add correct references for El Chichon.

Line 460: Chipperfield et al. used chemical transport model simulations. Âă Eyring et al., 2010 would be the correct reference.

References

1. Revisiting the hemispheric asymmetry in midlatitude ozone changes following the Mount Pinatubo eruption: A 3‐D model study SS Dhomse, MP Chipperfield, W Feng , GW Mann, ML Santee Geophysical research letters 42 (8), 3038-3047

2. An unexpected disruption of the atmospheric quasi-biennial oscillation Scott M. Osprey, Neal Butchart, Jeff R. Knight, Adam A. Scaife, Kevin Hamilton, James A. Anstey, Verena Schenzinger, Chunxi Zhang, Science23 Sep 1424-1427

3 Eyring et al., assessment of stratospheric ozone return dates and ozone recovery in CCMVal-2 models, Atmos. Chem. Phys., 10, 9451-9472, https://doi.org/10.5194/acp-10-9451-2010, 2010.

---

## Author Comment (AC1) · 8 Jan 2018

**Replies to reviewers of manuscript acp-2017-853**

Weber, M., Coldewey-Egbers, M., Fioletov, V. E., Frith, S. M., Wild, J. D., Burrows, J. P., Long, C. S., and Loyola, D.: Total ozone trends from 1979 to 2016 derived from five merged observational datasets – the emergence into ozone recovery, Atmos. Chem. Phys. Discuss., in review, 2017.

We are very grateful for the very helpful comments from both reviewers, which helped improving our manuscript. In the following we address all the comments raised. The replies to both reviewers are in Sections 1 and 2. The Appendix contains the revised manuscript in track mode.

**1. Reply to Reviewer #1**

**Reviewer comment (italics):**

(1) Please clarify the extent to which there are trends in any of the terms used in the MLR, whether there are uncertainties in those, and whether these in turn can influence the calculated ozone trends and their uncertainties. For example, I would argue that we do not know the trends in eddy heat fluxes in the stratosphere very well (although we may know the year to year variability, we do not know the longer term trends on decadal or multi-decadal time scales). So one question is: are there trends in the eddy heat flux terms that characterize the BDC components in the MLR? How uncertain are those trends? Could they (or do they) then influence the ozone trends that are the primary subjects of interest here? Papers on the uncertainties and differences between ERA and MERRA might be a useful point of reference here, but only a starting point; I think quantitative analysis is needed. The same could be said for trends in the solar term, for example. I am concerned that these could considerably influence the conclusions drawn, and should be discussed and documented.

**Our reply:**

In our MLR the proxy terms are not detrended. We believe that this has negligible impact on the trends. A similar MLR analysis was done in Chipperfield et al. (2017) on the same SBUV MOD total ozone dataset, where all proxy terms were detrended. The results of Chipperfield et al. (2017) are very close to the results in this study. Some of the proxy terms in Chipperfield et al. are also different from this study, e.g. QBO derived from a principal component analysis of the tropical stratospheric winds (in our study Singapore winds). In this study we attempt to isolate all possible factors in order to left the unknown (low frequency) changes in the linear trend terms that we then interpret as due to ODS and climate changes.

(2) Please clarify which terms in the MLR regression could conceivably involve feedbacks. For example, it is possible that changes in ozone play a role in the strength of the BDC (and this could happen not only on longer time scales, but also interannually). Has this been considered? Could it be important? If, for example, part of the BDC trend term is caused by ozone changes, then is your calculation of the 'ozone trend' potentially in error? By how much?

In an ideal world all factors (and proxy terms) are independent, but most dynamical processes are somehow related, e.g. ENSO, QBO, BDC, AO/AAO. For instance the annual mean QBO proxy has a correlation with the BDC on the order of 0.75. Since we mainly use the dynamical terms to remove the high frequency variability, the correlations are not really an issue. However, the interpretation of the individual dynamical factors may become more difficult but this is not the main goal here as our focus is on the trends. We report here trends using different regression equations, the standard which includes only QBO and ENSO and secondly the full regression that adds BDC and AO terms. The trend results are nearly identical, however, the uncertainty are slightly larger for the former.

(3) There are many studies providing evidence that the Antarctic ozone hole has influenced the strength of the southern annular mode (AO) in some seasons. Here is another potential feedback. The same questions apply as in item 2) above.

See our comment to (2). Generally, for the extra terms beyond the standard regression we only used them if their contribution was significant beyond  $2\sigma$ . In the Antarctic most of the large variability was

covered by the BDC. In the paragraph explaining the neglect of AAO, we however added the following text (line 255ff in the discussion paper): "... even though the Antarctic Oscillation (AAO), the counterpart of the AO in the NH, provides an important ozone feedback mechanism and is strongly

related to the Antarctic ozone hole (e.g. Thompson and Solomon, 2002), in this analysis this term is not robust as its significance strongly depends on whether the BDCs term is added or not."

(4) The fit to the interannual ups and downs in the ozone time series is pretty good, implying that on an interannual basis the terms involving dynamical variability are fairly well captured. Thus, the ups and downs are certainly not random noise – they are due to known and characterizable phenomena. The paper ought to discuss this, and make reference to the work of Shepherd et al., NatGeo, 2014, who combined a model with data to improve on the analysis. Based on the Shepherd paper, it does not seem reasonable to allow these variations to inflate the uncertainties on the ozone trend terms. Instead of doing the approach of combined MLR, would it not be more consistent to remove them first, and then examine the trends in the remainder. Terms involving inter-annual dynamical variations could conceivably be removed by detrending your index, and then regressing the detrended series to the ozone time series, and then doing MLR with the remaining terms you have. How would this or a similar approach affect the ozone trends, and in particular, their uncertainties? If this could significantly reduce the uncertainties in ozone trends (as I suspect, and as I think Shepherd et al. support), then that should at a minimum be stated since uncertainties are a key emphasis of the paper, and we need to know how robust the ozone trend uncertainties really are.

See also our replies above. The dynamical variability is real, known, and well captured by the regression models as well as by chemistry-climate(transport) models as shown from the ensemble means of the CCMVAL2 models (Eyring et al. 2010) and from the study by Shepherd et al. (2014) and Chipperfield et al. (2014). We consider the trend uncertainties reported in our study as lower limits since we still neglect uncertainties in the observations as well as in the merging to create the long-term timeseries (see Conclusion in our paper), Subtracting first all proxy terms from the observational timeseries and then determine the trends may reduce the uncertainties further. However, if uncertainties of the observations as well as the added uncertainties due to the pre-fitted proxy terms in the left hand side of such a regression equation (observation minus proxy terms) are properly accounted for (weighted least squares), I would guess that uncertainties in the trends would remain the same by fittiing with and without subtracting dynamical terms from the observations. We added the following to the Summary and Conclusions: "We may therefore conclude that we are about to emerge into the phase of ozone recovery as is also shown by chemistry-climate and chemistry-transport models (e.g. Eyring et al., 2010; Shepherd et al., 2014; Solomon et al., 2016; Chipperfield et al., 2017). Both the regression model (e.g. in our study) and models capture the dynamical variability well and their results are consistent."

(5) While the ILT and PLT trend approaches have certain advantages, they are allowed to float independent of each other. So what they lack is a grounding in the physics and chemistry that must link the processes that deplete ozone to those that make it recover. The advantage of EESC is that it has that grounding. The ILT 'advantage' over the PLT as suggested by your figure has a lot to do with the choice of year for the separation, which is a little arbitrary since there are uncertainties in EESC, and it does vary with height as well as latitude. I think the advantages and disadvantages of each approach should be discussed more clearly.

In Section 3.2 we discuss in detail the advantages and disadvantages in the choice of trend terms (ILT vs PLT vs. EEESC). It is true that the evolution ozone observations is largely consistent with the evolution of the EESC as confirmed by comparisons with the range and mean of the ensemble of climate models as discussed in Pawson et al. (2014). I agree that the exact form of the EESC curve is highly uncertain and their latitude and altitude dependence is not well known and has not been accounted for in any regressions so far. We added the following text to Section 3.2: "In the last WMO ozone assessment (Pawson et al., 2014) the evolution of total ozone was reported to be largely consistent with the range given by the ensemble of climate models accounting for ODS changes... The exact shape of the EESC curve as a function of altitude and latitude is highly uncertain. In most regressions only one representative EESC curve for the extratropics and polar regions, respectively, is fitted as calculated from tropospheric emissions assuming a certain age-of-air distribution (Newman et al., 2007)."

(6) I'm concerned about some of the statements regarding the aerosol fits, for several reasons. a) First, it's not obvious that the total SAD is the best predictor, because the distribution of aerosol and the distribution of ozone losses need not coincide. At a minimum, the paper should test what happens if the SAD amount at 70 mb is used instead. b) Second, the statement that you used the Mills et al. aerosols but don't get significant correlation with polar ozone (suggesting a conflict with Solomon et al., 2016) misses some key points. As indicated in Solomon et al., 2016 and discussed in greater detail in Ivy et al. (GRL, 2016), the dependence of aerosols on ozone loss will depend upon temperature; in a warm year even a big volcano doesn't have a big effect so the dynamics is important in setting the stage. To capture this, you might need some kind of mixed predictor including aerosols and temperature, but that's not what you used so it's misleading to say that you don't get a high correlation. c) Third, the key parameter emphasized in Solomon et al., 2016 was the area of the ozone hole, which is not the variable you have evaluated here.

These are good points and they reveal the limitations of an MLR analysis focussing only on observational data (without the aid of model results like in Solomon et al. and Shepherd et al.). We use the SAOD column amounts from Mills as we deal here only with ozone column observations. The Mills aerosol proxy is limited to the time after 1990 and the proxy fit is dominated by the Pinatubo period. As already noted each volcanic event may involve a different response (see discussion on differences of El Chichon and Pinatubo responses). Thus it may be quite difficult to isolate all minor events with stratospheric impact using several (and independent) aerosol proxy terms.

Stratospheric temperature is highly correlated with ozone changes (see Ball et al., 2017), however, almost all factors considered in the MLR affect temperature as well and complicates interpretation as temperature is not a "process." We add the following text in the polar trend section: "Even though we used the aerosol data from Mills et al. as used by Solomon et al. and Ivy et al. as input to their climate model, as a proxy in our regression, the impact of the aerosol term was found to be negligible in 2015. The apparent contradiction of the aerosol impact on Antarctic ozone between Solomon and Ivy et al. and our study should not be overstated here. The fitting of the aerosol proxy data based upon Mills et al. is dominated by the Pinatubo event and may therefore not be properly scaled during the Cabuco volcanic event. It is difficult to isolate minor volcanic events with stratospheric impact in the MLR using separate aerosol proxy terms as is done for the larger El Chichon and Pinatubo events. This is clearly a limitation of the MLR approach."

**(7) Page 2, line 44 what is the reference for this number for changes in ODS levels?**

The rate of decrease in ODS relative to earlier increase (1/3) was determined from linear fitting the EESC timeseries (outside the polar regions) before and after the turnaround point (Dhomse et al. (2006). The rate of 1/3 is only valid for EESC curves assuming a mean age of air of 3 years. We added the following "see Fig. 2 in Dhomse et al. (2016)"

**(8) Page 3, line 73. Similar but not the same? What is different from Chehade? Please summarize what you changed, and why**

We added in the manuscript the following: "The main difference to the earlier study is that we use in this paper five merged datasets while in Chehade et al. (2014) only the GSG and SBUV MOD datasets were used. All datasets used here were updated up to and including 2016 (four more years added). In Chehade et al. (2014) the piecewise linear trends (PLT) and EESC term were fitted, while here only the independent linear trends (ILT) before and after the turnaround in ODS are considered for the reasons discussed in Section 3.2."

**2. Reply to Reviewer #2**

*Line 13: other*  $\rightarrow$  *others*

done

line 18: Do you mean "dynamical variability"

add "dynamical"

line 22: "outside the tropics ozone profile peaks

no changes

line 164: please mention the number of outliers in percentage

It is difficult to specify the exact number, we instead refer to Figs. 10 and 11 in Coldewey-Egbers et al. (2015) in the text.

*Line 191-195- please mention the data sets with largest biases for few latitude bands along with absolute biases In DU.*

As mentioned in the text the trends calculated are identical to those calculated for ozone anomalies. The biases between the datasets do not play a role here.

*Line 224: "deep total ozone minimum-> large ozone decrease*

Done

*Line 228: confusing. those two references are about Pinatubo eruption. Also, it is not only dynamics but vertical distribution of SAD caused increase in SH mid- stratospheric ozone (e.g. Dhomse et al., 2015)*

Both Aquila and Schnadt-Poberaj discuss the hemispheric asymmetry in the observed ozone effect. The Dhomse paper has been added.

*Line 228: SAD= area density but SAOD= stratospheric aerosol optical depth*

Change to SAOD

Line 229: Can't understand why you are using WACCM simulated SAD, which is still data. I think CMIP6 recommended SAD data or AOD@550nm) until 2014 would have been much more realistic ftp://iacftp.ethz.ch/pub\_read/. For remaining two years, WACCM-based or any other data would be OK.

We did not use SAD, but SAOD at 550 nm from the WACCM model. The impact from the recent smaller eruption was rather small (compared to the Pinatubo effect).

*Line 250: what about tropics? Did you try to use mean heat flux from both the hemispheres (BDCn +BDCs)/2.?*

The eddy heat fluxes were calculated separately for each hemisphere (between 45 and 75 deg latitudes). They were never combined into a kind of "global" BDC index in this study (and we are not aware if anybody has done this). We added a small clarification: "For each day the area weighted mean of the 100 hPa eddy heat flux between 45 and 75 latitudes separately for each hemisphere is calculated and the monthly mean timeseries derived …" (added: separately for each hemisphere).

*Line 280: reword "roughly equally contributed"*

Changed to "roughly contributed half each"

line 289: Volcanically enhanced stratospheric aerosol

Done

line 310: with an uncertainty just less than (delete of)

Done

line 314: I think "Osprey et al, 2016" would be a better reference.

We added this reference, however, the Tweedy paper is more appropriate as it focuses on the ozone impact.

line 382: Again. What happens if you combine EP fluxes from both the hemispheres?

In an initial regression runs all terms incl. BDCs and BDCn were included for all latitudes. The significance of the BDC term from one hemisphere to the other hemisphere was always statistically insignificant. Except for the terms in the standard equations, extra terms were only used if their contributions were significant.

Line 397: Do you Mt. Pinatubo eruption? Or add correct references for El Chichon.

Added the Dhomse et al. (2015) paper (see reply earlier).

*Line 460: Chipperfield et al. used chemical transport model simulations. Eyring et al., 2010 would be the correct reference.*

Added the Eyring et al. reference

**Appendix**

Revised manuscript in track mode

**Total ozone trends from 1979 to 2016 derived from five merged observational datasets - the emergence into ozone recovery**

Mark Weber1, Melanie Coldewey-Egbers2, Vitali E. Fioletov3, Stacey M. Frith4, Jeannette D. Wild5,6, John P. Burrows1, Craig S. Long5, and Diego Loyola2

1University of Bremen, Bremen, Germany
 2German Aerospace Center (DLR), Oberpfaffenhofen, Germany
 3Environment and Climate Change Canada, Toronto, Canada
 4Science Systems and Applications Inc., Lanham, MD, USA

[revised manuscript text omitted]

•

| zonal bands                     | MLR      |                            | NASA    | NOAA    | GSG     | GTO     | WOUDC   |
|---------------------------------|----------|----------------------------|---------|---------|---------|---------|---------|
| $-60^{\circ}$ S $-60^{\circ}$ N | full     | trend >1996 [%/dec.]       | +0.2(3) | +0.5(4) | +0.7(3) | +0.2(3) | +0.6(3) |
| annual                          |          | trend $\leq$ 1996 [%/dec.] | -1.8(7) | -2.0(7) |         | —       | -1.2(6) |
|                                 |          | $r^2$                      | 0.92    | 0.92    | 0.94    | 0.94    | 0.92    |
|                                 |          | $\chi$ [DU]                | 1.3     | 1.3     | 1.2     | 1.2     | 1.2     |
| -60°S-60°N                      | standard | trend >1996 [%/dec.]       | +0.2(3) | +0.5(3) | +0.7(3) | +0.2(3) | +0.6(4) |
| annual                          |          | trend $\leq$ 1996 [%/dec.] | -2.1(7) | -2.3(7) |         |         | -1.7(6) |
|                                 |          | $r^2$                      | 0.90    | 0.91    | 0.91    | 0.93    | 0.86    |
|                                 |          | $\chi$ [DU]                | 1.3     | 1.4     | 1.3     | 1.2     | 1.4     |

bold numbers: statistical significance at  $2\sigma$

.

**Figure 1.** Illustration of different choices of trend terms commonly used in MLR applied to total ozone. Blue: EESC (effective equivalent stratospheric chlorine); Red: piecewise linear trends before and after  $t_0 = 1996$  (PLT) also called hockey-stick; Green: independent linear trends (ILT). Black curve shows the NH total ozone timeseries from NOAA SBUV V8.6. The red dotted line indicates that the PLT is mathematically equivalent to using a trend change term in the MLR. The injection point is the point where the trend change terms starts, here in year 1996. All fits were done using only the linear regression terms in Eq.1 or, alternatively, the EESC curve replacing linear regression terms. See discussion in main text.